# Metal to non-metal sites of metallic sulfides switching products from CO to $CH_4$ for photocatalytic $CO_2$ reduction

Yao Chai[1,4], Yuehua Kong [2,4], Min Lin[1], Wei Lin [2], Jinni Shen[1], Jinlin Long [1], Rusheng Yuan[1], Wenxin Dai [1,3], Xuxu Wang [1] & Zizhong Zhang [1,3] ✉

The active center for the adsorption and activation of carbon dioxide plays a vital role in the conversion and product selectivity of photocatalytic $CO_2$ reduction. Here, we find multiple metal sulfides $CuInSnS_4$ octahedral nanocrystal with exposed (1 1 1) plane for the selectively photocatalytic $CO_2$ reduction to methane. Still, the product is switched to carbon monoxide on the corresponding individual metal sulfides $In_2S_3$, $SnS_2$, and $Cu_2S$. Unlike the common metal or defects as active sites, the non-metal sulfur atom in $CuInSnS_4$ is revealed to be the adsorption center for responding to the selectivity of $CH_4$ products. The carbon atom of $CO_2$ adsorbed on the electron-poor sulfur atom of $CuInSnS_4$ is favorable for stabilizing the intermediates and thus promotes the conversion of $CO_2$ to $CH_4$. Both the activity and selectivity of $CH_4$ products over the pristine $CuInSnS_4$ nanocrystal can be further improved by the modification of with various co-catalysts to enhance the separation of the photogenerated charge carrier. This work provides a non-metal active site to determine the conversion and selectivity of photocatalytic $CO_2$ reduction.

The solar-energy-driven photocatalytic conversion of $CO_2$ with $H_2O$ into hydrocarbon fuels is a significant solution for simultaneously addressing global energy demands and climate change issues[1–4]. Various products of $CO_2$ reduction from photocatalytic multi-electron processes, including CO (two electrons), HCOOH (two electrons), HCHO (four electrons), $CH_3OH$ (six electrons), and $CH_4$ (eight electrons), have been produced by a great variety of photocatalysts[5–8]. Achieving both high selectivity and high conversion for photocatalytic $CO_2$ reduction is highly desirable in the field of photocatalysis research. However, efficient photoreduction of $CO_2$ is very challenging, both in terms of chemical thermodynamics and kinetics, due to the highly stable structure of $CO_2$ and the involvement of multiple proton-coupled electron transfer[9–11]. Additionally, the regulation of product selectivity in photocatalytic $CO_2$ conversion remains an unknown challenge.

It has been well understood for the photocatalytic process that the identification of the active centers of catalysts for the adsorption and activation of $CO_2$ is prerequisite for efficient $CO_2$ conversion and product selectivity. Constructing an active center of catalysts for the adsorption and activation of $CO_2$ is an efficient solution to improve $CO_2$ conversion efficiency and product selectivity[12–14]. Various metal-free photocatalysts were reported for $CO_2$ reduction[15–19], typically such as covalent organic frameworks, graphitic carbon nitride, elemental phosphorus, boron nitride, and silicon carbide[20–24]. These metal-free photocatalysts have non-metallic sites as the adsorption and activation sites of $CO_2$ molecules and thus photocatalytical $CO_2$ reduction[20,25–28]. However, for metal oxide or sulfide photocatalysts, many studies suggest that metal components or defects on photocatalysts play a crucial role as primary sites in the adsorption and activation of $CO_2$ and thus affect product selectivity[29]. Zhou et al. reported that the S vacancy

[1]State Key Lab of Photocatalysis on Energy and Environment, College of Chemistry, Fuzhou University, Fuzhou, P. R. China. [2]College of Chemistry, Fuzhou University, Fuzhou, P. R. China. [3]Qingyuan Innovation Laboratory, Quanzhou, P. R. China. [4]These authors contributed equally: Yao Chai, Yuehua Kong. ✉e-mail: z.zhang@fzu.edu.cn

or Cd vacancy CdS with single Au atom deposition for $CO_2$ adsorption is different[30]. $CO_2$ prefers to physically adsorb on single Au atoms of $Au/CdS_{1-x}$ and photoreduction into CO, while $CO_2$ is more likely to chemically bond on the Cd vacancies of $Au/Cd_{1-x}S$, resulting in a remarkable CO and $CH_4$ generation rate on $Au/Cd_{1-x}S$. He et al. synthesized a $ZnIn_2S_4$ nanosheet photocatalyst with abundant Zn vacancies[31], where $CO_2$ can be efficiently adsorbed on Zn vacancies to form $CO_2^-$ species and highly selective photoreduction into CO. Yu et al. designed a $Cu_3SnS_4$ photocatalyst with S vacancies to increase ratios of Cu (I/II) for $CO_2$ photoreduction[32]. The formed Cu (I) acts as adsorption sites for $CO_2$, conducive to further hydrogenation of CO intermediate into $CH_4$. Xie et al. showed that the defect-state $CuIn_5S_8$ ultrathin nanosheets have low-coordination Cu and In sites for $CO_2$ adsorption to form highly stable Cu-C-O-In intermediates, which tend to obtain 100% $CH_4$ selectivity[33]. Xu et al. designed a Co-Ni-P NH/BP catalyst with bimetallic sites to form a highly stable Co-O-C-Ni intermediate for the selective photoreduction of $CO_2$ to $CH_4$[34]. However, the complex structures of defects on photocatalyst make it only a plausible correlation between defect structures and product selectivity. Some research shows that the adsorbed interaction between $CO_2$ and metal sites is relatively weak since the formed metal-C bonds are weaker than the highly stable C=O bonds in $CO_2$. This leads to the easy cleaving of metal-C bonds during the reaction process, hindering the deep reduction of $CO_2$ into hydrocarbons[30]. Obviously, the non-metal sites on metal sulfide photocatalysts are very rarely considered the primary active center for the adsorption and activation of $CO_2$.

Here, we have successfully prepared multiple metal sulfides, including $CuInSnS_4$ octahedral nanocrystal and corresponding individual metal sulfides $In_2S_3$, $SnS_2$, and $Cu_2S$, through a simple one-step hydrothermal method. The $CuInSnS_4$ nanocrystal is thermodynamically favorable to activate $CO_2$ and leads to a switch of main products from CO to $CH_4$ with a yield of 6.53 μL h⁻¹ for the visible-light-driven $CO_2$

reduction with $H_2O$ vapor without the assistance of any noble metal cocatalysts. In contrast, individual metal sulfides can only produce CO. We reveal that different adsorption configurations of $CO_2$ on metal sulfides lead to different products in $CO_2$ photoreduction. The non-metal sulfur atom in the prepared multiple metal sulfides $CuInSnS_4$ octahedron nanocrystal is thermodynamically favorable to activate $CO_2$ and leads to a switch of main products to $CH_4$, as compared with the common individual metal sulfides $In_2S_3$, $SnS_2$ and $Cu_2S$ with metal center as active sites to form CO products. $CO_2$ is revealed to be adsorbed on the S atom center of $CuInSnS_4$ to form an S-C-O-In structural unit, which is more conducive to protonation and leads to the efficient photocatalytic yield of $CH_4$. Thus, we provide an insight into the role of non-metal center of photocatalyst in determining the conversion and selectivity of photocatalytic $CO_2$ reduction. Although the pristine $CuInSnS_4$ only exhibits a yield of $CH_4$ evolution of 6.53 μL h⁻¹ (corresponding to 5.83 μmol h⁻¹ g⁻¹), the activity and selectivity of $CH_4$ evolution on $CuInSnS_4$ can be significantly improved by modifying with cocatalysts such as Pt, CoO, NiO, and $Co(OH)_2$. We believe that this knowledge can contribute to the development of more efficient and selective photocatalysts for $CO_2$ reduction in the future.

## Results
### Characterization of $CuInSnS_4$ octahedral nanocrystal
The as-prepared $CuInSnS_4$ nanocrystals belong to the cubic spinel structure (JCPDS No. 29-0548), as revealed by the X-ray diffraction pattern (Fig. 1a), which were prepared through a simple one-step hydrothermal reaction. The hydrothermal temperature (160 °C, 180 °C, and 200 °C) did not have any evident impact on the crystalline and purity of $CuInSnS_4$ (Supplementary Fig. 1). Under similar hydrothermal processes, $In_2S_3$ with a tetragonal crystal phase structure (JCPDS No. 25-0390), $SnS_2$ with a hexagonal crystal phase (JCPDS No. 23-0677), and $Cu_2S$ with a cubic crystal phase were also prepared (JCPDS No. 02-1284)

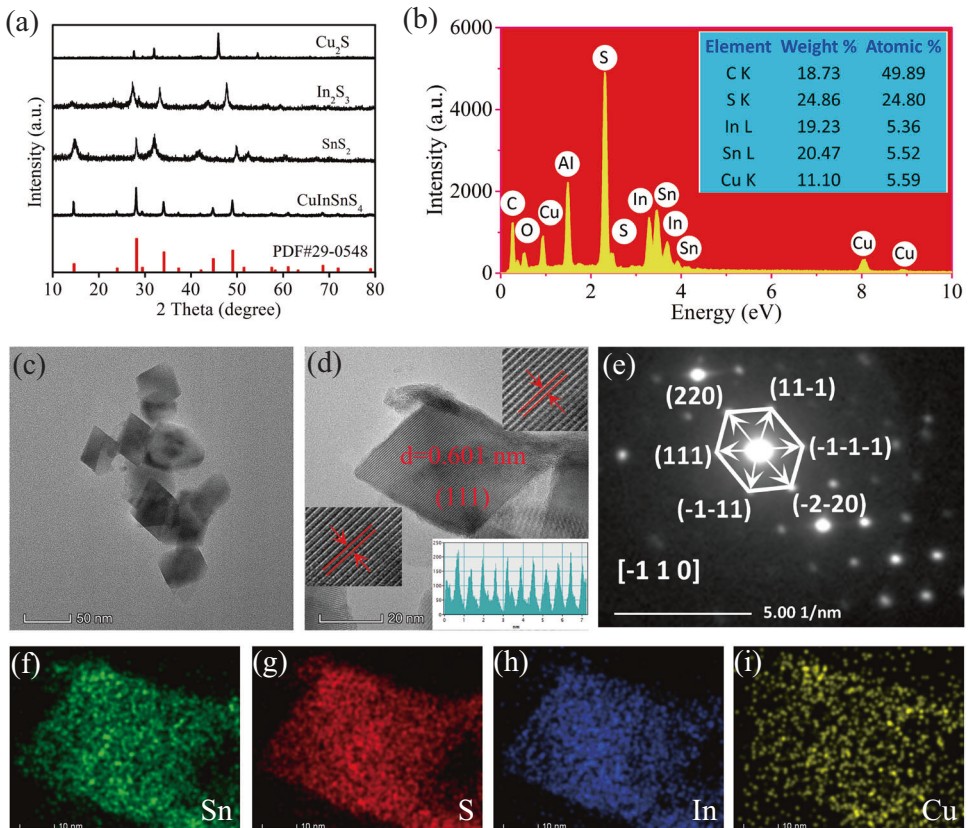

**Fig. 1 | The characterization of crystal phase, composition, and morphology. a** XRD patterns of several metal sulfides, **b** EDS spectra of $CuInSnS_4$ photocatalyst, **c** a TEM image of $CuInSnS_4$ sample, **d** an HRTEM image, and **e** an SAED pattern of the $CuInSnS_4$ sample, as well as **f–i** EDS elemental mapping images.

(Supplementary Fig. 2). The cubic spinel CuInSnS$_4$ crystal structure shows that either In or Sn atom is coordinated with six S atoms to form an octahedron structure, while Cu atom is formed by a [CuS$_4$] tetrahedral structural unit (Supplementary Fig. 3). For comparison, in In$_2$S$_3$ crystals, the In atom is present in [InS$_4$] tetrahedron and [InS$_6$] octahedron coordination (Supplementary Fig. 4). Cu and Sn atoms exist as [CuS$_4$] tetrahedron and [SnS$_6$] octahedron coordination in Cu$_2$S and SnS$_2$ crystal structure, respectively (Supplementary Figs. 5, 6). The coordination structure of each metal in CuInSnS$_4$ coincides with the individual Cu$_2$S, In$_2$S$_3$, and SnS$_2$. The composition of CuInSnS$_4$ is analyzed by energy-dispersive spectroscopy, as shown in Fig. 1b. EDS offers that the Cu, Sn, In, and S atomic ratio is ~1:1:1:4, very close to the stoichiometric value of CuInSnS$_4$ compounds, indicating the high purity of CuInSnS$_4$ nanocrystal. Meantime, the ICP-MS test results of the metal element content of the nano-single crystal CuInSnS$_4$ sample are presented in Table S1. The atomic ratio of Cu:In:Sn is 1.06:1.00:1.00, which closely matches the theoretical value of CuInSnS$_4$. Supplementary Fig. 7 shows the SEM images of In$_2$S$_3$, Cu$_2$S, and SnS$_2$ samples. In$_2$S$_3$ exhibits a morphology of microspheres self-assembled from nanosheets. Cu$_2$S has the basic shape of nanoparticles, while SnS$_2$ displays the morphology of ultrathin nanosheets. Both scanning electron microscopy and transmission electron microscopy images of nanoscale microstructure confirm that the prepared CuInSnS$_4$ displays an octahedral nanocrystal structure with a size of about 30 nm (Supplementary Fig. 8 and Fig. 1c). The high-resolution TEM image (Fig. 1d) shows the exposed facets of the octahedron with a lattice spacing of 0.601 nm, which is assigned to (111) facets of CuInSnS$_4$. Selected area electron diffraction further verifies that the CuInSnS$_4$ sample not only exposes the (1 1 1) crystal plane but also has a single crystal structure (Fig. 1e). All the results confirm the successful preparation of CuInSnS$_4$ nanocrystal with high-quality exposed (1 1 1) crystal faces. The specific surface area of the CuInSnS$_4$ sample is about 24.1 m$^2$ g$^{-1}$, while that of the prepared In$_2$S$_3$, SnS$_2$, and Cu$_2$S are 26.8 m$^2$ g$^{-1}$, 37.6 m$^2$ g$^{-1}$, and 3.6 m$^2$ g$^{-1}$, respectively (Supplementary Fig. 9). As displayed in Fig. 1f−i, the uniform distribution of Sn, S, In, and Cu elements in CuInSnS$_4$ octahedral nanoparticles indicates that the catalyst is of high purity.

X-ray photoelectron spectroscopy was used to compare the electronic states of the obtained sample. The Cu$2p_{3/2}$ and Cu$2p_{1/2}$ binding energies of CuInSnS$_4$ sample are 932.07 eV and 951.90 eV, respectively (Fig. 2a). This demonstrates that the valence state of Cu is +1 in the CuInSnS$_4$ sample[35,36], which is also confirmed by the Cu LMM spectra (Supplementary Fig. 10). Notably, the Cu$2p$-binding energies of the CuInSnS$_4$ sample is identical to that of Cu$_2$S. The binding energies of In$3d_{5/2}$ and In$3d_{3/2}$ in the CuInSnS$_4$ sample are 444.63 eV and 452.18 eV, respectively. These values indicate that the valence state of In in the CuInSnS$_4$ sample is +3. Compared with In$_2$S$_3$, the In$3d$ binding energy of CuInSnS$_4$ uniformly shifts toward the lower binding energy, as shown in Fig. 2b. This is attributed to the difference in the In coordinated environment between CuInSnS$_4$ and In$_2$S$_3$ because the partial In atom in In$_2$S$_3$ exists in the state of [InS$_4$] tetrahedron[37]. In the CuInSnS$_4$ sample, the Sn$3d_{5/2}$ and Sn$3d_{3/2}$ doublets are centered respectively at 486.30 eV and 494.70 eV, assigning to Sn$^{4+}$ valence state. Notably, the binding energy of Sn in CuInSnS$_4$ is slightly lower than that in the parent SnS$_2$ (Fig. 2c). The possible reason is that the Sn atoms are in different crystal structures[38]. Furthermore, the binding energies of S$2p_{3/2}$ and S$2p_{1/2}$ in the CuInSnS$_4$ sample are measured to be 161.45 eV and 162.70 eV, respectively, which corresponds to the S$^{2-}$ valence state. In the S$2p$ XPS spectra, the binding energies of S atoms increase in the order of Cu$_2$S < In$_2$S$_3$ < SnS$_2$ < CuInSnS$_4$, as shown in Fig. 2d. S atoms in CuInSnS$_4$ have the highest binding energy. This is interpreted by the fact that the average bond length (0.253 nm) between sulfur and metal atoms in CuInSnS$_4$ is slightly larger than in monometallic sulfide[4,37,39]. Moreover, the S atom in CuInSnS$_4$ has a higher binding energy than that of monometallic sulfides, indicating an electron-deficient state of the S atoms in CuInSnS$_4$ compared to monometallic sulfides. This electron-deficient state of the S atom in CuInSnS$_4$ can serve as the reaction site for CO$_2$ adsorption and activation.

## Photocatalytic conversion of CO$_2$ and H$_2$O vapor

The photocatalytic CO$_2$ reduction performance of the samples was evaluated in a customized sealed quartz glass vessel, in a gas-solid reaction system, with a small amount of water vapor in a CO$_2$

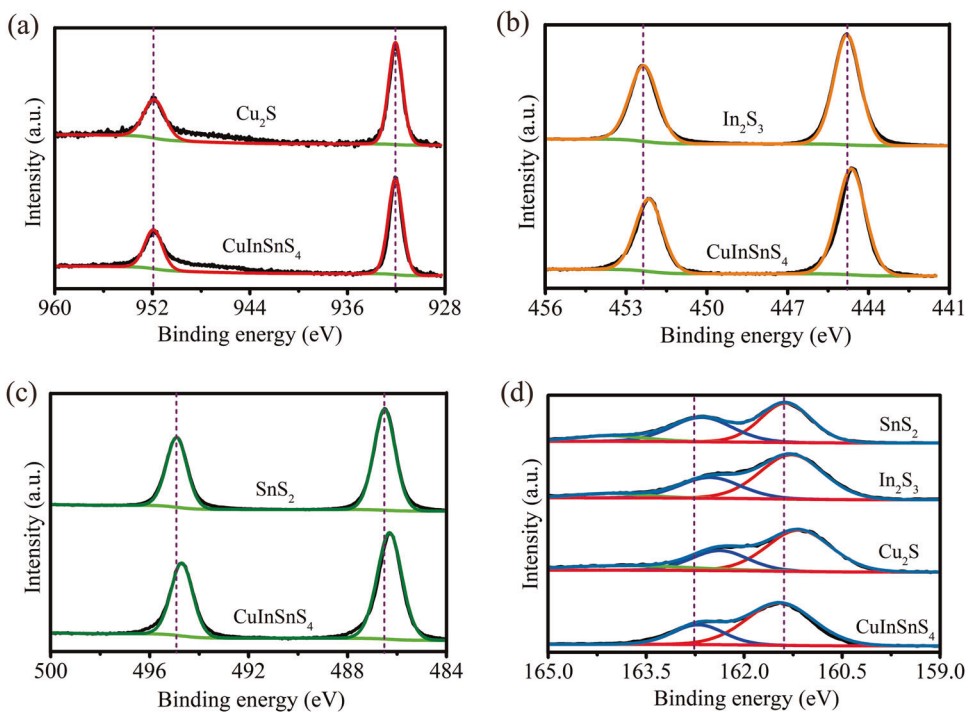

**Fig. 2 | Characterization of electronic structure.** High-resolution XPS spectra of metal sulfides: **a** Cu$2p$, **b** In$3d$, **c** Sn$3d$, and **d** S$2p$.

atmosphere, under the irradiation of a 300 W Xe lamp with a 420 nm cutoff wavelength filter (Supplementary Fig. 11). It is important to note that this reaction is a gas-solid phase reaction. The system contains only 50 µL of water, which evaporates into water vapor upon injection into the reactor. As a result, only gaseous products such as $CH_4$, CO, and a small amount of $H_2$ are detected, while liquid products (such as $CH_3OH$, HCHO, HCOOH, etc.) are not detected. Figure 3a shows the corresponding photocatalytic $CO_2$ performance of the samples under visible-light irradiation. For single metal sulfides $In_2S_3$, $Cu_2S$, and $SnS_2$, CO is the main product with a yield rate of less than 3.2 µL h$^{-1}$ from photocatalytic $CO_2$ reduction, while the multi-electron transfer product $CH_4$ is hardly formed. Whereas, the $CuInSnS_4$ sample shows excellent photocatalytic $CO_2$ reduction performance, yielding $CH_4$ as the main product besides a slight amount of CO and $H_2$ evolution. The hydrothermal temperature has no obvious impact on the $CO_2$ reduction performance of $CuInSnS_4$ (Supplementary Fig. 12). The rate of $CH_4$ generation reaches 6.53 µL h$^{-1}$ for the $CuInSnS_4$ sample. The selectivity of $CH_4$ is calculated to be 67.3% based on the contents of carbon-containing products. The significant difference in the product selectivity demonstrates the different mechanisms of $CO_2$ reduction or the different active sites between $CuInSnS_4$ and single metal sulfides. The controlled blank experiments under other conditions were investigated to confirm the occurrence of $CO_2$ reduction on $CuInSnS_4$, as shown in Supplementary Fig. 13. The $CH_4$ product is not detected without either light irradiation or catalyst, proving that $CO_2$ reduction is a light-induced catalytic reaction on $CuInSnS_4$. Meanwhile, without adding $H_2O$ into the reaction system, only a very few products are detected, indicating that $H_2O$ is also one of the essential reactants involved in the reaction. When $N_2$ is used instead of $CO_2$ for the reaction, only a small amount of CO is detected, directly proving that the source of CO and $CH_4$ products is $CO_2$. The presence of small amounts of CO may be attributed to contamination from ambient air, reactors, and equipment, as we can see that small CO products is also

detected without $CuInSnS_4$ photocatalysts[40]. Figure 3b shows the stability of the $CuInSnS_4$ sample for photocatalytic reduction of $CO_2$. It is clear that the $CuInSnS_4$ sample presents good performance without noticeable activity decrement after three-cycle photocatalytic $CO_2$ reduction tests of a total of 27 h (9 h visible-light irradiation for each cycle). Neither crystal structural transformation nor absorption behavior changes are found in the XRD pattern and ultraviolet–visible diffuse reflectance spectra for the $CuInSnS_4$ sample after photocatalytic reaction (Supplementary Fig. 14). These results suggest that $CuInSnS_4$ possesses good stability during photocatalytic $CO_2$ reduction. Moreover, the XPS of the catalyst after the reaction shows that a weak photocorrosion phenomenon occurs in $CuInSnS_4$[41,42] (Supplementary Fig. 15). Specifically, the photogenerated holes or the active oxygen species oxidize the surface $S^{2-}$ of the catalyst to $SO_3^{2-}$. The peaks with binding energies in the range of 168.26-170.26 eV are assigned to the XPS peaks of $SO_3^{2-}$ species (Supplementary Fig. 16). The photocatalytic performance of $CuInSnS_4$ sample is evaluated under different monochromatic light wavelengths in Fig. 3c. It is observed that as the wavelength of the incident light increases, the photocatalytic activity significantly decreases. However, the $CuInSnS_4$ nano-single crystal photocatalyst demonstrates a $CH_4$ generation rate of 0.69 µL h$^{-1}$ and CO generation rate of 0.22 µL h$^{-1}$ under the irradiation of 525 nm monochromatic light, which is surprising. The different monochromatic light tests indicate that $CuInSnS_4$ is an exceptional catalyst for $CO_2$ photoreduction under visible light. The $^{13}CO_2$ isotope experiment further validates that $CH_4$ product is generated from the photoreduction of $CO_2$ molecules, where only $^{13}CH_4$ is detected when the reaction is carried out in a $^{13}CO_2$ atmosphere, as shown in Fig. 3d. Meanwhile, the $^{13}CO_2$ isotope also confirms that the generated CO was indeed a product of $CO_2$ photoreduction. When the reaction is conducted in a $^{13}CO_2$ atmosphere, the weak peak of $^{13}CO$ with a m/z = 29 was detected due to the low activity of $CuInSnS_4$ for CO evolution (Supplementary Fig. 17a). It is noteworthy that the mass spectrum peak

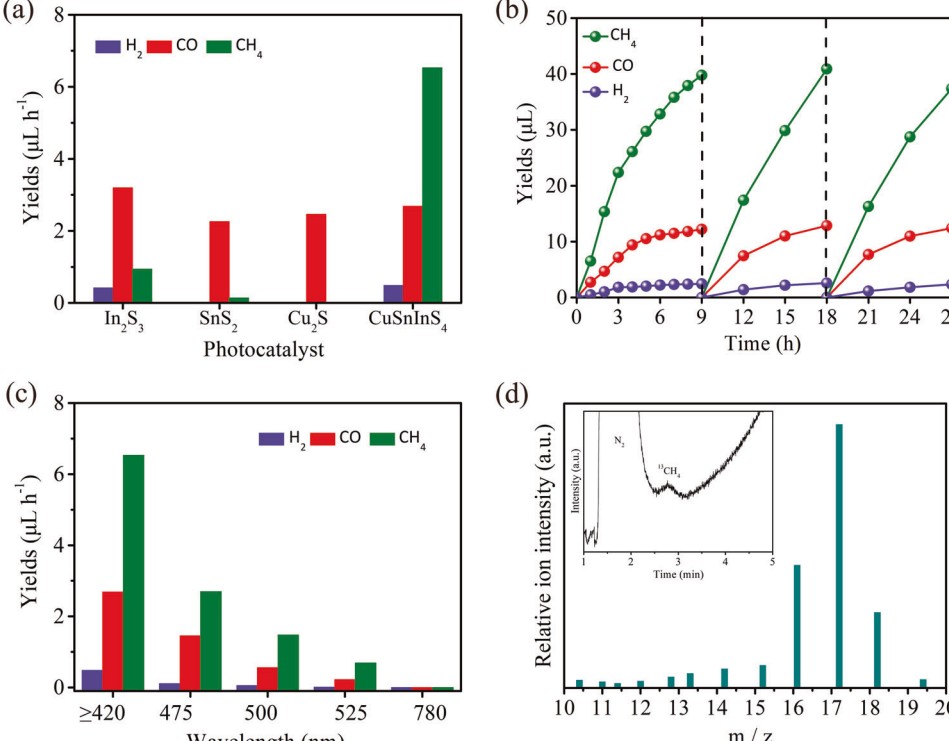

**Fig. 3 | Photocatalytic conversion performance of $CO_2$ and $H_2O$. a** The production rate of $CH_4$, CO, and $H_2$ in various photocatalysts under visible-light conditions. **b** Photocatalytic $CO_2$ reduction stability test of $CuInSnS_4$ sample. **c** Photocatalytic performance of $CuInSnS_4$ sample under monochromatic light irradiation. **d** GC-MS spectra of $^{13}CH_4$ generated from $^{13}CO_2$.

at m/z = 28 corresponds to the $N_2$ molecule from air interference, as evident from the distinct $N_2$ peaks in chromatogram. Additionally, the $^{12}CO_2$ experiment further evidences that the generated CO results from the photoreduction of $CO_2$ molecules with no peaks at m/z = 29 (Supplementary Fig. 17b).

Although the pristine $CuInSnS_4$ only exhibits a yield of $CH_4$ evolution of 6.53 µL h$^{-1}$ (corresponding to 5.83 µmol h$^{-1}$ g$^{-1}$) with a selectivity of 67.3%, the activity and selectivity of $CH_4$ evolution on $CuInSnS_4$ can be improved by coupling with semiconductor photocatalysts or noble metals as cocatalysts. We evaluated the photocatalytic performance of $CuInSnS_4$ modified with Pt, CoO, NiO, $Co(OH)_2$, and dual co-catalysts Pt and $Co(OH)_2$, for the $CO_2$ reduction reaction. The composition and chemical states of Pt, CoO, NiO, $Co(OH)_2$ cocatalysts are well verified by XRD patterns and XPS spectra (Supplementary Figs. 18–23). Table S2 lists a comparison of the photoreduction activity of $CuInSnS_4$ and co-catalyst-modified $CuInSnS_4$ photocatalysts, along with the common metal sulfide systems currently used for $CO_2$ photoreduction. Clearly, both the yield and selectivity of $CH_4$ evolution on $CuInSnS_4$ can be significantly improved by modifying with cocatalysts such as Pt, CoO, NiO, and $Co(OH)_2$. The activity of $CH_4$ evolution on the modified $CuInSnS_4$ photocatalysts surpasses the majority of the reported photocatalysts for $CO_2$ reduction up to now. Particularly, the incorporation of $Co(OH)_2$ as a cocatalyst significantly enhances the $CO_2$ photoreduction activity of the $CuInSnS_4$ photocatalyst. As the $Co(OH)_2$ loading increases, the photoreduction activity of $CO_2$ exhibits a characteristic volcanic pattern. With 5%$Co(OH)_2$ loading onto $CuInSnS_4$, the production rates for $CH_4$ and CO, respectively, reach 145.45 and 32.32 µmol h$^{-1}$ g$^{-1}$, corresponding to a $CH_4$ selectivity of 81.8%. The generation rates of $CH_4$ and CO are 25 times and 13 times that of pure $CuInSnS_4$, respectively. Furthermore, when $CuInSnS_4$ is modified with a dual co-catalyst of 5% $Co(OH)_2$ as an oxidation co-catalyst and 1%Pt as a reduction co-catalyst, $CH_4$ production reaches 195.60 µmol h$^{-1}$ g$^{-1}$, along with 22.00 µmol h$^{-1}$ g$^{-1}$ of CO, and a $CH_4$ selectivity of 89.9%. Photoelectrochemical characterization was employed to assess the separation efficiency of photogenerated carriers on the modified $CuInSnS_4$, as shown in Supplementary Fig. 24. Clearly, the $CuInSnS_4$ samples modified with the co-catalyst exhibit a higher photocurrent signal and a smaller electrochemical impedance radius as compared to the parent $CuInSnS_4$ sample. The photocurrent increases sequentially in the order of $CuInSnS_4$ < 5%$Co(OH)_2$/$CuInSnS_4$ ≈ 1%Pt/$CuInSnS_4$ < 5%$Co(OH)_2$/$CuInSnS_4$/1%Pt, indicating that the modification of the dual co-catalyst improves photoelectric carrier separation and migration compared to the single co-catalyst. The NiO and CoO cocatalysts also improve the separation efficiency and migration rate of the photogenerated carriers of $CuInSnS_4$ photocatalyst. The decreasing order of electrochemical resistance radius is $CuInSnS_4$ > 1%Pt/$CuInSnS_4$ ≈ 5%$Co(OH)_2$/$CuInSnS_4$ > 5%$Co(OH)_2$/$CuInSnS_4$/1%Pt, consistent with the photocatalytic activity trend. Additionally, both 10%NiO/$CuInSnS_4$ and 10% CoO/$CuInSnS_4$ exhibit smaller electrochemical resistance radii than pure $CuInSnS_4$, confirming that the cocatalyst promotes the photogenerated charge separation and migration. Therefore, the modification of $CuInSnS_4$ with various cocatalysts to enhance the separation of photogenerated carriers is related to the activity and selectivity of $CH_4$ products. The apparent quantum yield is calculated by measuring the yield of $CH_4$ and CO in 5%$Co(OH)_2$/$CuInSnS_4$/1%Pt under monochromatic light at 400 nm. Supplementary Fig. 25 shows the spectrum and intensity of monochromatic light at 400 nm. Under 400 nm monochromatic light irradiation, the apparent quantum efficiencies for $CH_4$ and CO are 0.16% and 0.01%, respectively. Based on the above analysis, we believe that $CuInSnS_4$ nano-single crystal photocatalysts through further optimization design of the different contents and types of cocatalyst modification can be more efficient and selective for $CO_2$ reduction in the future.

## Energy band and photoelectrochemical characterization

The band energy potential is a key determinant of the driving force of redox reactions. Therefore, we have studied the band structure of the catalyst through UV-vis DRS and XPS valence band spectroscopy. As shown in Fig. 4a and Supplementary Fig. 26, the optical absorption band edge of $CuInSnS_4$ is calculated to be 787.5 nm, which corresponds to a band gap energy of 1.57 eV. For comparison, the absorption band edges of single metal sulfide $Cu_2S$, $SnS_2$, and $In_2S_3$ are 747.0 nm, 552.7 nm, and 641.6 nm, corresponding to the band gap of 1.66 eV, 2.24 eV, and 1.93 eV, respectively. Moreover, the valence band potential of $CuInSnS_4$ is determined to be 0.50 V from the valence band XPS spectra (Supplementary Fig. 27), while the $Cu_2S$, $SnS_2$, and $In_2S_3$ possess valence band potentials of 1.02 V, 1.70 V, and 1.93 V, respectively. By using the formula $E_{CB} = E_g - E_{VB}$, we have determined that the conduction band potentials of $CuInSnS_4$, $Cu_2S$, $SnS_2$, and $In_2S_3$ are −1.15 V, −1.10 V, −0.54 V, and −0.55 V, respectively. Based on the optical band gaps, we have obtained the electronic band energies relative to a normal hydrogen electrode (Fig. 4b), indicating that both $CuInSnS_4$ and single metal sulfides have the ability to reduce $CO_2$ to $CH_4$ and CO. Notably, $CuInSnS_4$ exhibits the highest reduction potential for the photogenerated electrons to reduce $CO_2$. Additionally, $CuInSnS_4$ shows a significantly increased photocurrent density (0.014 mA cm$^{-2}$) compared to the single metal sulfide under visible-light irradiation (Fig. 4c), indicating a more efficient separation of the photoinduced charge in multi-metal sulfides. The lower interface resistance in the corresponding electrochemical impedance spectra (Fig. 4d) confirms the rapid transfer of photogenerated electrons in $CuInSnS_4$. The efficient separation efficiency and migration rate of photogenerated carriers make polymetallic sulfides exhibit higher photocatalytic $CO_2$ reduction performance compared to monometallic sulfides. However, the higher migration and separation efficiency of charge carriers in $CuInSnS_4$ alone is not sufficient to explain the substantial difference in the product selectivity between $CuInSnS_4$ and single metal sulfides.

## The in situ $CO_2$ adsorption FT-IR spectra and mechanism

To understand the $CO_2$ reduction process over $CuInSnS_4$ and single metal sulfides, in situ Fourier-transform infrared spectroscopy is used to compare the reaction intermediates on the catalyst surface. No macroscopic infrared absorption peaks for intermediates are found on $Cu_2S$ or $SnS_2$, even under light irradiation, possibly due to their weak chemical interaction with $CO_2$ (Supplementary Fig. 28a, b). However, $In_2S_3$ shows a significant activation effect on the $CO_2$ adsorbed on the surface under light irradiation (Fig. 5a). Notably, $CO_2$ can form chemical adsorption with $In_2S_3$ even under dark conditions, as indicated by the infrared peak at 1150 cm$^{-1}$, which can be considered an O-S stretching vibration[43], suggesting that the oxygen atom of $CO_2$ is bonded to the sulfur atom of $In_2S_3$. Upon light irradiation, some infrared peaks of the produced intermediates on the catalyst surface are observed. The infrared peak at 1225 cm$^{-1}$ is attributed to the vibration of bidentate bicarbonate[44], while the infrared peak at 1412 cm$^{-1}$ is attributed to the vibration of monodentate bicarbonate[45]. Most importantly, the infrared peak at 1610 cm$^{-1}$ is attributed to the *COOH group, which is generally regarded as the crucial intermediate for $CO_2$ reduction to CO[46]. Surprisingly, the polymetallic sulfide $CuInSnS_4$ exhibits strong chemisorption of $CO_2$ and strong physisorption of $H_2O$ (Fig. 5b). The prominent infrared peak observed at 1627 cm$^{-1}$ is attributed to the physical adsorption of $H_2O$. However, $In_2S_3$ does not exhibit a noticeable infrared adsorption peak of $H_2O$ at 1627 cm$^{-1}$. This is because $In_2S_3$ has a lower affinity towards water adsorption than $CuInSnS_4$, as indicated by the high contact angle on $In_2S_3$ surface (Supplementary Fig. 29). Moreover, a prominent infrared peak at 1117 cm$^{-1}$ assigned to C-S stretching vibration is observed upon the $CO_2$ adsorption on $CuInSnS_4$[47–49]. The $CO_2$ adsorption and

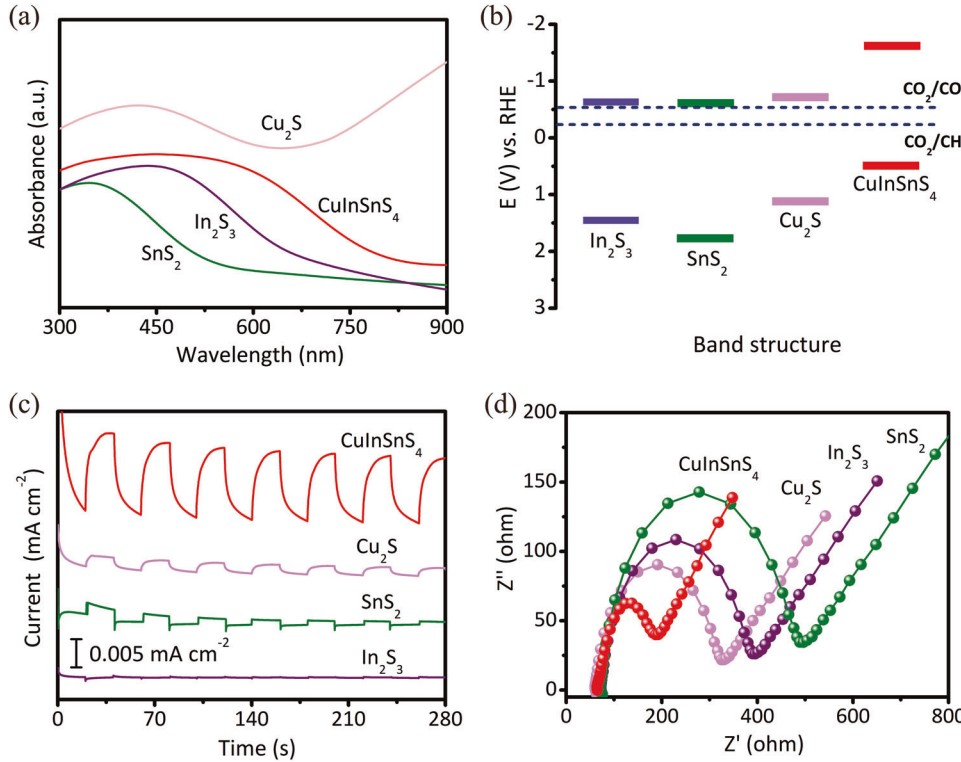

**Fig. 4 | Characterization of energy bands and optoelectronic properties. a** UV-vis absorption spectrum of various metal sulfides. **b** The optical band gap energy ($E_g$) of the corresponding CuInSnS$_4$ and various single metal sulfides. **c** Photocurrent response and **d** electrochemical impedance spectroscopy of the as-prepared samples.

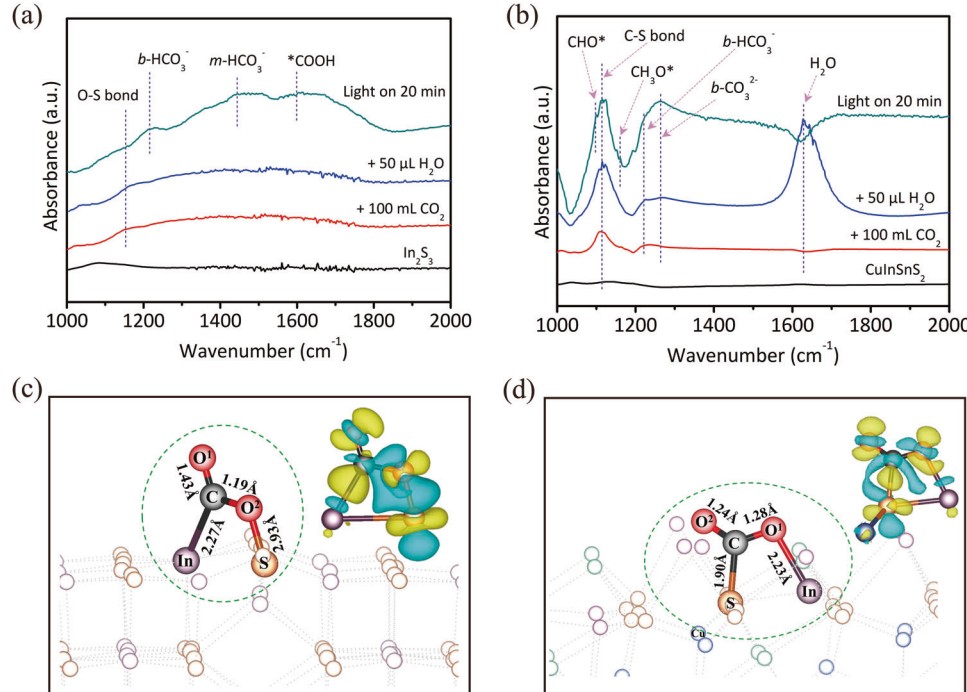

**Fig. 5 | In situ $CO_2$ adsorption FT-IR spectra and differential charge density map as well as free energy diagrams of $CO_2$ photoreduction to $CH_4$.** In situ FT-IR spectra of $CO_2$ adsorbed on **a** In$_2$S$_3$ and **b** CuInSnS$_4$. $CO_2$ adsorption configuration and differential charge density map of **c** In$_2$S$_3$ and **d** CuSnInS$_4$ photocatalysts. In the differential charge density map, the yellow and blue regions indicate electron accumulation and depletion, respectively.

activation are significantly improved on $CuInSnS_4$ compared to $In_2S_3$. Furthermore, the adsorption state of $CO_2$ on $CuInSnS_4$ is different from that on $In_2S_3$, as the carbon atom of $CO_2$ is bonded to the sulfur atom of $CuInSnS_4$. Under light conditions, $CuInSnS_4$ produces specific $CO_2$-activated intermediates, as indicated by the infrared peaks at $1225$ $cm^{-1}$ and $1260$ $cm^{-1}$, attributed to the vibration of bidentate bicarbonate and bidentate carbonate[50,51], respectively. Additionally, the infrared peaks at $1100$ $cm^{-1}$ and $1160$ $cm^{-1}$ are attributed to the absorption peaks of *CHO and $*CH_3O$, which are intermediates for the yield of $CH_4$[52,53]. Therefore, either the $CO_2$ adsorption state in darkness or the produced intermediates under light irradiation show that the sites for $CO_2$ adsorption and the $CO_2$ reduction approach differ between single-metal sulfide $In_2S_3$ and multi-metal sulfide $CuInSnS_4$. This may account for the different selectivity of products between $In_2S_3$ and $CuInSnS_4$.

It has been demonstrated that sulfur defect sites in multiple metal sulfides acted as an active center for the selective photoreduction of $CO_2$ to $CH_4$. However, $CuInSnS_4$ nano-single crystal shows no significant sulfur defect signals compared to the strong defect signals in $SnS_2$ and $In_2S_3$, as shown in Supplementary Fig. 30. This indicates that the selectivity of $CH_4$ products on $CuInSnS_4$ is not related to sulfur defects. The mechanism of selective photocatalytic $CO_2$ reduction on $CuInSnS_4$ and $In_2S_3$ photocatalysts is further theoretically studied. Firstly, we investigated the adsorption behavior of $CO_2$ on the surfaces of $In_2S_3$ and $CuInSnS_4$. The (0 0 1) crystal plane of the $In_2S_3$ sample and the (1 1 1) crystal plane of the $CuInSnS_4$ sample were selected as models, and all atoms on the crystal planes were considered as potential sites for $CO_2$ adsorption activation (Supplementary Figs. 31 and 32). Based on $CO_2$ adsorption energy, the optimal adsorption models of $CO_2$ on $In_2S_3$ and $CuInSnS_4$ photocatalyst surfaces are optimized (Supplementary Figs. 33 and 34). Figure 5c, d depict schematic diagrams of $CO_2$ stable adsorption configurations and the charge density difference of $CO_2$ on $In_2S_3$ and $CuInSnS_4$, respectively. The stable $CO_2$ adsorption configuration on $In_2S_3$ is the C atom of $CO_2$ bonded to In atom with a bond length of $2.27$ Å, while the O atom of the $CO_2$ molecule is bonded with the S atom with a bond length of $1.70$ Å. The $CO_2$ adsorption model for the polymetallic sulfide $CuInSnS_4$ is the opposite. The unique C-S bond with a bond length of $1.90$ Å is formed between the C atom of $CO_2$ and the surface S atom, and one O atom of

$CO_2$ is bonded with In atom with a bond length of $2.23$ Å. The various adsorption configurations of $CO_2$ on $In_2S_3$ and $CuInSnS_4$ surfaces are attributed to the coordination environment and charge number of the surface S atoms. The surface S atom of $In_2S_3$ is an electron-rich site with $[SIn_3]$ coordination structure, while the S atom on $CuInSnS_4$ is an electron-poor center with $[SInSnCu]$ coordination structure (Supplementary Fig. 35 and Table S1). Different adsorption configurations may be the key to determining the direction of electron transfer and thus the selectivity of $CO_2$ reduction on $CuInSnS_4$ and $In_2S_3$. Adsorption of $CO_2$ on the $In_2S_3$ (0 0 1) crystal plane leads to inconsistent changes in two C=O lengths. The length of the $C-O^2$ bond is $1.43$ Å, equal to the ordinary C-O ($1.43$ Å) single bond, while the length of $C-O^1$ is shortened to $1.19$ Å, close to the length of C-O ($1.12$ Å) in a CO molecule. This asymmetric activation is more likely to cause the rupture of $C-O^2$, thereby preferentially producing CO on $In_2S_3$. In the case of the $CuInSnS_4$ (1 1 1) crystal plane, both C-O bonds are similar in length, measuring $1.26 \pm 0.02$ Å. They are longer than the C-O bond ($1.16$ Å) in a free $CO_2$ molecule, indicating that the bond energy of two C=O bonds of the activated $CO_2$ is simultaneously weakened. The calculation of the charge density difference reveals the difference in electronic structure and electron flow resulting from the interaction of $CO_2$ with the surface atoms of $In_2S_3$ and $CuInSnS_4$. On the $In_2S_3$ surface, there is extensive charge depletion for the $C-O^2$ and $S-O^2$ bonds, which implies that these chemical bonds are weakened. In contrast, $CO_2$ exhibits a wide charge accumulation region on $CuInSnS_4$, leading to the formation of a strong C-S and In-O chemical bond. This strong interaction is beneficial for the firm adsorption of $CO_2$ on the $CuInSnS_4$ surface, promoting the further deep reduction reaction. The Bader charges analysis further confirms that there is more charge transfer between the $CuInSnS_4$ surface and $CO_2$ molecules adsorbed on it. The surface of $In_2S_3$ and $CuInSnS_4$ loses $0.28e$ and $0.32e$, respectively, after $CO_2$ adsorption (Table S3).

The stable configuration of $CO_2$ adsorption on $CuInSnS_4$ determines its excellent $CO_2$ photoreduction activity and selectivity. Therefore, DFT calculations were further carried out to study the conversion pathway of $CO_2$ on the $CuInSnS_4$ photocatalyst surface, as shown in Fig. 6. In Fig. 6a, the adsorption configuration of $CuInSnS_4$ is shown for each intermediate step, from $CO_2$ adsorption to $CH_4$ generation. The C atoms of various intermediates, such as $CO_2*$, COOH*,

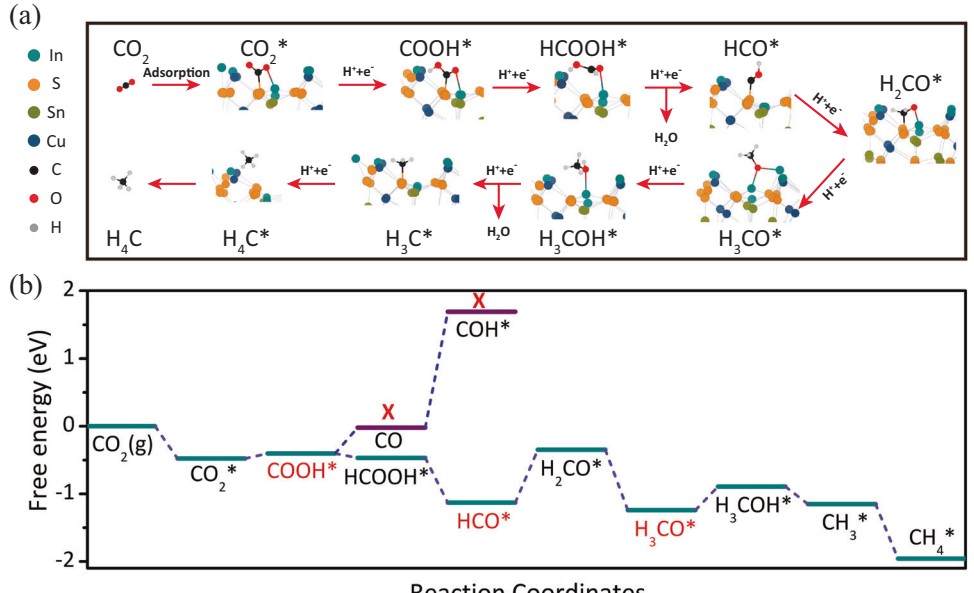

**Fig. 6 | DFT calculations of the $CO_2$ conversion pathway. a** Calculated adsorption configuration of $CO_2$ and reactive intermediates on CuSnInS$_4$. **b** Gibbs free energy diagrams for $CO_2$ reduction to $CH_4$ on CuSnInS$_4$.

CHO*, CH$_2$O*, and CH$_3$*, remain in a stable bond to the electron-deficient S atoms of the CuInSnS$_4$ nano-single crystal (111) plane. With the addition of protons and electrons, the removal of the H$_2$O molecule results in the breaking of the chemical bond between O atoms of intermediates and In atoms on CuInSnS$_4$. Moreover, unlike most metal sulfide photocatalysts, the hydrogenation of CO$_2$ adsorbed on the CuInSnS$_4$ surface to COOH* is an easy step, requiring only a potential energy barrier of 0.075 eV[33], as shown in Fig. 6b. This is attributed to the fact that the electrons are localized on the O$^2$ atom of CO$_2$ molecules in the S-C-O-In adsorption configuration of CO$_2$ on the CuInSnS$_4$ surface to facilitate the addition of protons, thus lowering the formation energy of COOH* intermediates. However, converting COOH* to CO is an endothermic reaction that must overcome an energy barrier of 0.46 eV. On the other hand, the continuous hydrogenation of COOH* intermediates to produce HCOOH* is an exothermic reaction, promoting the hydrogenation of CO$_2$. The formation of H$_2$CO* intermediates is the rate-limiting step for further hydrogenation processes, but the hydrogenation of H$_2$CO* to H$_3$CO* and finally to CH$_4$ is thermodynamically spontaneous. Therefore, CuInSnS$_4$ can achieve high selectivity for CH$_4$ products. Additionally, the adsorption energies of each intermediate product can explain the high CH$_4$ selectivity. Supplementary Fig. 36 shows that the adsorption energies of CH$_4$, CH$_3$OH, and HCOOH on the CuInSnS$_4$ surface are −0.17, −0.61, and −0.67 eV, respectively, with CH$_4$ having the highest adsorption energy. This indicates that CH$_4$ products are most easily desorbed from the CuInSnS$_4$ surface, which is one of the reasons why the CuInSnS$_4$ photocatalyst has high selectivity for the photoreduction of CO$_2$ into CH$_4$.

The photoreduction mechanism of CO$_2$ on the CuSnInS$_4$ surface is proposed in Fig. 7. The first step involves CO$_2$ adsorbing on the catalyst surface to form the unique S-C-O-In structural unit. This process weakens the C=O double bond in the CO$_2$ molecule while non-metallic S atoms serve as adsorption sites, ensuring a strong bond to the C atom of CO$_2$. This benefits the continuous reduction of CO$_2$ molecules into COOH*, HCO*, H$_3$CO* intermediates, and ultimately into CH$_4$ on the catalyst surface through the assistance of photogenerated electrons and protons. Lastly, the low adsorption energy of CH$_4$ on the catalyst surface facilitates its quick release, completing the full photocatalytic cycle reaction.

In summary, a CuSnInS$_4$ nano-single-crystal photocatalyst with exposed (1 1 1) facets is successfully prepared by a simple one-step hydrothermal reaction. Under visible-light irradiation, the CuSnInS$_4$ nano-single crystal photocatalyzes the conversion of CO$_2$ and H$_2$O into main products of CH$_4$ with a generation rate of 6.53 μL h$^{-1}$, significantly higher than that of single metal sulfides (In$_2$S$_3$, Cu$_2$S, and SnS$_2$). The electron-poor center sulfur atom on the CuSnInS$_4$ (111) crystal plane acts as the site for CO$_2$ adsorption and activation, which leads to the activation of the two symmetrical C=O double bonds of CO$_2$ molecule to form a stable S-C-O-In transition state. This induces CH$_4$ generation via the conversion route of COOH*→HCOOH*→H$_2$CO*→H$_3$CO*→CH$_4$*. However, the asymmetric activation of CO$_2$ by monometallic sulfides is more likely to result in the cleavage of individual C-O bonds in the CO$_2$ molecule, leading to the preferential photoreduction of CO$_2$ to CO. This work provides a distinctive understanding of catalysts for CO$_2$ adsorption and activation for the CO$_2$ selective conversion to help the conversion of CO$_2$ resources into high-value-added products.

## Methods

### Preparation of CuInSnS$_4$ nanocrystal

A simple one-step hydrothermal method was used to synthesize CuInSnS$_4$ nanocrystal photocatalyst with the cubic crystal structure. The detailed operation process is as follows. Firstly, 1 mmol of CuCl, 1 mmol of SnCl$_4$·5H$_2$O, and 1 mmol of InCl$_3$·4H$_2$O were added to 40 mL of deionized water to form a solution under vigorous stirring. Then, 5 mmol of TAA was dissolved in the above-mixed solution and stirred at room temperature for 30 min. Finally, the resulting mixed solution was transferred to a 50 mL Teflon-lined autoclave and sealed into a stainless steel tank for hydrothermal reaction. The hydrothermal temperature is controlled at 160, 180, and 200 °C for 24 hours. After the reaction, the product was collected and washed with deionized water, and dried under vacuum at 60 °C. The obtained samples were labeled CuInSnS$_4$ (160 °C), CuInSnS$_4$ (180 °C), and CuInSnS$_4$ (200 °C) according to the reaction temperature. The detailed preparation processes of CuInSnS$_4$ modified Pt, CoO, NiO, and Co(OH)$_2$ cocatalysts can be seen in supplementary information.

### Characterization

X-ray diffractometer (D8 Advance, Bruker) was used to analyze the crystal structure of the catalyst. The XRD test range is 10°–80°, and the scan rate is 10° min$^{-1}$. Scanning electron microscopy (su8010, Hitachi) was used to observe the surface morphology of the catalyst. The element composition and ratio of the sample are detected by EDS. The apparent morphology and high-resolution TEM image of the catalyst were tested by transmission electron microscope (TEM, TECNAI G2F20, FEI Company). At the same time, SAED and element mapping images of the catalyst were obtained in the TEM measurement mode. A UV-VIS-NIR Spectrophotometer (Cary 500) was used to obtain the catalyst UV-VIS-NIR DRS, in which BaSO$_4$ was used as a standard sample for 100% light transmission. The Micromeritics 3020 M physical adsorption instrument was used to obtain the nitrogen adsorption and desorption curves of different catalysts. The specific surface area of each catalyst was calculated from the type of nitrogen adsorption and desorption curves. The catalyst and dried potassium bromide were evenly ground, and 20 mg was weighed and pressed into slices, then placed in a quartz infrared tube for a carbon dioxide adsorption infrared test. In situ infrared spectra measurements were performed using a Fourier-transform infrared spectrometer (Nicolet iS50 FT-IR Spectrometer) equipped with a mercury cadmium telluride detector (Supplementary Fig. 37). In situ infrared spectra were recorded by averaging 32 scans at a resolution of 4 cm$^{-1}$. To initiate the experiment, the catalyst was placed in a 250 mL quartz tube and compacted into a film. The tube was then subjected to vacuum treatment for 60 min. Subsequently, high-purity CO$_2$ gas was introduced, and the quartz tube was sealed. A liquid sampler was used to inject 60 μL of deionized water into the sealed quartz tube. The tube was heated with a hot blower to vaporize the

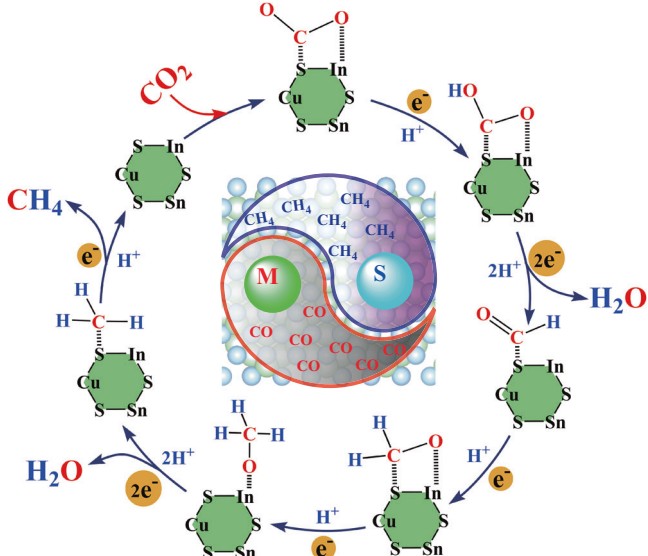

**Fig. 7 | CO$_2$ photoreduction pathway.** Proposed photocatalytic mechanism for CO$_2$ reduction on the CuInSnS$_4$. The backgroup crystal structure was created by VESTA program[56].

deionized water. The quartz tube was positioned in the FT-IR spectrometer, ensuring that the incident light of the infrared spectrometer was perpendicular to the sample surface. A xenon lamp visible-light source was introduced to directly illuminate the sample surface. Infrared spectra were recorded after pretreating the catalyst in a vacuum, introducing $CO_2$ gas, and vaporizing deionized water, respectively. After the introduction of light, infrared spectra were recorded every 5 min. Gas chromatography-mass spectrometry (Agilent 7890B, Agilent 5977B MSD) was used to analyze $^{13}CH_4$ and $^{13}CO$. Electron paramagnetic resonance spectroscopic measurements were performed at room temperature using a Bruker A300 EPR spectrometer.

## Photocatalytic performance

300 W xenon lamp (Microsolar 300, Beijing Perfectlight Technology Co., Ltd.) was equipped with a 420 nm cutoff wavelength filter as a light source that simulates visible light for photocatalytic $CO_2$ reduction tests. Firstly, 50 mg of catalyst was dispersed in 5 mL of deionized water and sonicated for 10 min. Then the catalyst was dropped into a watch glass with a diameter of 3 cm and dried at 80 °C. Subsequently, the dried catalyst was placed in a quartz reactor with a volume of ~250 cm³, and then high-purity $CO_2$ gas (99.999%) was introduced to replace all the air. The flow rate of carbon dioxide gas was ~100 mL/min and lasted for 1 h. Finally, 50 μL deionized water was injected into the quartz reactor from the rubber stopper through a gas chromatography liquid syringe (maximum range, 50 μL), and the reactor was heated with a hair dryer to evaporate the water into water vapor. As a result, water is present in the form of water vapor throughout the reaction. The reactor was placed under a xenon lamp for photocatalytic reaction, and the current of the xenon lamp was 16 A. After 1 h of illumination, 0.5 mL of gas was extracted from the reactor with a 1 mL gas chromatograph gas syringe and injected into the gas chromatograph for product analysis and detection. Among them, $H_2$, $O_2$, and $N_2$ are detected by a thermal conductivity detector. $CH_4$ was detected by the flame ionization detector. CO passes through the flame ionization detector after being transformed by the nickel reformer. The product was qualitatively and quantitatively analyzed by gas chromatography retention time and appearance standard curve method.

## Theoretical calculations

The density functional theory calculations were performed using the VASP code with the projected augmented wave method[54]. Generalized gradient approximation in the scheme of Perdew-Bueke-Ernzerhof was used for the exchange-correlation functional[54]. The PBEsol exchange-correlation function was adopted in the optimization calculations. Grimme's DFT-D3 scheme was used to describe the long-range vdW interactions[54]. The cutoff energy of the basis function was 420 eV. For the $CuSnInS_4$ (1 1 1) crystal plane, a $2 \times 2 \times 1$ supercell with a four-layer slab was constructed, and only the top two layers were allowed to relax. A vacuum region of 12 Å was set above the surface for periodic boundary conditions, and dipole correction was also applied. Gamma-centered $1 \times 1 \times 1$ grid k-points were used for the interface. Geometry relaxation was performed until the energy, and atomic forces converged to be smaller than $10^{-5}$ eV and 0.03 eV/Å. Charge transfers were calculated using the Bader charge analysis[54,55].

The free energy of each reaction intermediate was determined by: $G = E + ZPE - TS$. The electronic energy was directly obtained from DFT calculations. The zero-point energy and entropy correction (TS, T = 298.15 K) were computed from vibration analysis according to standard methods. The adsorption-free energy of the adsorbates can be calculated using: $\Delta G_{ads} = \Delta E_{ads} + \Delta ZPE - T\Delta S$, where $\Delta E_{ads}$ is the adsorption energy of the adsorbates, and $\Delta ZPE$ and $\Delta S$ are the difference between ZPE and S, respectively. After the adsorption-free energies of the adsorbates are obtained, the reaction-free energies of $CO_2$ reduction reaction elementary steps can be determined correspondingly by using the computational hydrogen electrode model[40].

## Data availability

The authors declare that the data supporting the findings of this study are available in the paper and its supplementary information files. Source data are provided with this paper.

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

## Acknowledgements

This work is financially supported by the National Natural Science Foundation of China (Grants no. 21972020, Z.Z.), and the Natural Science Foundation of Fujian Province of P. R. China (2020L3003, Z.Z.).

## Author contributions

X.W. and Z.Z. conceived the concept and supervised the research. Y.K., W.L., and J.S. performed the DFT calculations. Y.C. and M.L. performed the experimental work. J.L., R.Y., and W.D. helped to write—review & edit the manuscript. All authors designed the experiments, analyzed the data, and drafted the manuscript.

## Competing interests

The authors declare no competing interests.
