## [Peer Review File · Nature Communications]

Metal to non-metal sites of metallic sulfides switching products from CO to CH₄ for photocatalytic CO₂ reductionREVIEWER COMMENTS

Reviewer #1 (Remarks to the Author):

Zhang et al. reported CO₂ reduction activity of individual metal sulfides including In₂S₃, SnS₂, and Cu₂S and multi-metal sulfides (CuInSnS₄), and demonstrate the reduction products can be switched from CO to CH₄ by adjusting the metal to non-metal sites in metallic sulfides. Despite the detailed study of the mechanism, the inferior selectivity and poor yields severely limits such system for practical applications. As far as I know, conversions from CO₂ to CO or CO₂ to CH₄ are extensively studied, and much higher yield and selectively is reported, such as the following references [Chem. Commun., 2021, 57, 8468-8471; Appl. Catal., B, 2021, 285, 119773; Adv. Funct. Mater., 2021, 31, 2010780; ACS Sustain. Chem. Eng. 2019, 7, 650-659; J. Am. Chem. Soc. 2018, 140, 14595-14598]. The novelty of this manuscript is not enough to match the high-level journal, Nature Communications, so I strongly recommend to reject it.

The following issues need to be considered before publication on any journal.

1. From the introduction, we can see that one key concern of this manuscript is the selectivity of CO₂ reduction. As the author put forward, high production selectivity has always been the goal of CO₂ reduction. The design strategy of catalysts presented by the author is to achieve highly efficient and selective conversion of CO₂. However, according to the catalytic results provided by the authors, the multi-metal catalyst increases the variety of gas-phase products and significantly reduces the overall selectivity when compared to Cu₂S (100% CO selectivity) and SnS₂ (>90% CO selectivity). This goes against the original intention of designing high selectivity catalysts.
2. Various work on the role of non-metals as catalytic sites for CO₂ reduction has been reported, and they have achieved efficient/highly selective conversion of CO₂. The citation of relevant literature is severely insufficient.
3. EDS is not a reliable method for studying element content. It is recommended that the author implement ICP-MS to determine the content and proportion of metals.
4. For the control experiments, the production of CO was still observed under N₂ atmosphere. In the isotope labelling experiments, the authors only give the GC-MS spectra of the CH₄ product, without mentioning CO. It is recommended that the CO isotope labelling experiments be monitored and the relevant GC-MS spectra be provided.
5. The author is suggested to pay attention to the details of the article, for example in line 64 "CO₂".

Reviewer #2 (Remarks to the Author):

This manuscript investigated the effect of metallic and nonmetallic sites as CO₂ adsorption sites on the product selectivity and activity of photocatalytic CO₂ reduction. The results demonstrated that the electron-deficient central S atom on the crystal plane of multinary metal sulfide CuInSnS₄ (1 1 1) served as the CO₂ adsorption and activation site. A stable transition state [S-C-O-In] was formed to allow CH₄ to be efficiently generated through the conversion pathway of COOH* → HCOOH* → H₂CO* → H₃CO* → CH₄*. However, the asymmetric activation of CO₂ by monometallic sulfides (In₂S₃) was more likely to result in the cleavage of a single C-O bond in the CO₂ molecule. This led to the preferential photoreduction of CO₂ to CO. This work is of significant importance for the study of photocatalytic CO₂ conversion from the perspective of catalysis, including the adsorption sites of substrate molecules, adsorption configurations, energy barriers of reaction intermediates, desorption of products, and other related fields. Therefore, the manuscript can be accepted for publication after the following improvement.

Detailed recommendations are as follows:

1. The XPS tests indicated that the In 3d binding energy of CuInSnS₄ was uniformly shifted to a lower binding energy compared to In₂S₃. Moreover, the binding energy of S atoms increased in the order of Cu₂S < In₂S₃ < SnS₂ < CuInSnS₄. These results served as evidence that S atoms lacked electrons while In atoms gained electrons on the (1 1 1) crystal face of CuInSnS₄ nano single crystal. Similarly, on the In₂S₃ (0 0 1) crystal face, the S atoms were electron-rich, and the In atoms were electron-

deficient. Therefore, the author should provide a further analysis and discussion of the XPS results to support their findings.

2. The authors claimed that the EDS analysis showed the atomic ratio of Cu, Sn, In, and S to be approximately 1:1:1:4, which was said to be very close to the stoichiometric value of CuInSnS₄ compound, thus indicating high purity of CuInSnS₄ nanocrystals. However, this description was inadequate. The authors should provide specific atomic ratios of Cu, Sn, In, and S to accurately determine the proximity to the stoichiometric value of the CuInSnS₄ compound.

3. The addition of water resulted in a significant infrared peak of water in CuInSnS₄ nano-single crystals, as shown in Fig. 5b, whereas In₂S₃ did not exhibit an obvious water adsorption IR peak. why?

4. Why is this reaction considered a gas-solid reaction? It appears that during the photocatalytic CO₂ reduction test, 50 μL of deionized water was injected into the quartz reactor. The authors claimed that the deionized water was evaporated by heating it with a hair dryer. However, if the water exists in the form of water vapor during the reaction, it would mean that the reaction temperature is at least 100°C or higher. In reality, the test was conducted under normal temperature and pressure conditions. Therefore, the reaction should be classified as a gas-liquid-solid three-phase reaction.

5. The authors suggested that different coordination environments of S atoms on the In₂S₃ and CuInSnS₄ surfaces, as well as the difference in charge density on the S atoms, led to different adsorption configurations of CO₂, as shown in Supplementary Fig. 22 and Table S1. However, as seen from the schematic diagram of the crystal structure, the S sulfur atom of In₂S₃ was mainly coordinated with three In atoms, regardless of whether the crystal face was exposed or not, whereas the S sulfur atom of CuInSnS₄ was mainly coordinated with one In atom, one Cu atom, and one Sn atom. Since CuInSnS₄ mainly exposed the (1 1 1) crystal plane, the plane primarily distributed S and In atoms, whereas Cu and Sn atoms were located in deeper layers, rendering them unsuitable for CO₂ adsorption sites. Therefore, the exposure of different crystal planes, as well as the different coordination environments of S atoms and differences in charge density on S atoms, were the reasons for the different adsorption configurations of CO₂, not just the different coordination environments of surface S atoms and the differences in S atom charge densities.

6. The authors explained the differences in the photocatalytic CO₂ reduction activity between monometallic sulfides and CuInSnS₄ nano single crystals by carrier separation efficiency and transfer rate. Additionally, they attributed the difference in product selectivity to the difference in adsorption configuration. However, it was worth noting that carrier separation efficiency and mobility rate were also responsible for the difference in CO₂ photoreduction product selectivity since CH₄ (8-electron reduction product) and CO (2-electron reduction product) required different amounts of photogenerated electrons. Furthermore, the degree of activation of the catalyst for CO₂ adsorption can also contribute to the difference in CO₂ photoreduction activity since a strong adsorption and activation ability can reduce the energy barrier of the reaction and facilitate the efficient conversion of CO₂.

7. Additionally, (1) It was important to note that (1 1 1) was a crystal plane but not a surface. The author has erroneously referred to (1 1 1) as the surface of CuInSnS₄ in several instances throughout manuscript. (2) Photocurrent scale was missed in Fig 4c. (3) The authors should provide details about the light source used for photocatalytic CO₂ reduction, including light intensity and spectrum.

Reviewer #3 (Remarks to the Author):

In this manuscript, the authors reported that a variety of metal sulfide photocatalysts (In₂S₃, SnS₂, Cu₂S, CuInSnS₄) were synthesized using a simple hydrothermal method, and their application in the photocatalytic conversion of CO₂ with H₂O was investigated. The photocatalytic performance tests revealed that, compared to the single metal sulfides, the polymetallic sulfide CuInSnS₄ exhibited higher activity for CO₂ conversion and also showed a change in product selectivity from CO to CH₄. Through systematic characterization and testing, the authors elucidated the CO₂ conversion on the S atom sites of multimetal sulfide CuInSnS₄ different from the metal sites of common single metal sulfides. Overall, the manuscript was well-prepared and well-characterized, and it will be of interest to

researchers in the field of photocatalysis research. Therefore, I recommend publishing this work with appropriate modifications. Here are my comments and suggestions.

1. Please provide the standard XRD patterns of the single metal sulfide samples, including In_2S_3 , SnS_2 , and Cu_2S .
2. The polymetallic sulfide CuInSnS_4 exhibits octahedral nanoparticles with exposed (111) facets. However, it is not clear to the morphology characteristics of monometallic sulfides (In_2S_3 , SnS_2 , and Cu_2S). It is recommended that the authors supplement the SEM or TEM images of monometallic sulfides.
3. The authors should provide a more comprehensive analysis and discussion of the XPS results, including the determination of the chemical valence states of each element in the CuInSnS_4 sample before and after the reaction. This detailed characterization will allow for a better understanding of the CuInSnS_4 sample. Furthermore, by examining the change in the valence state of the catalyst before and after the reaction, the stability of the catalyst can be assessed. Hence, it is recommended to include a detailed analysis of the valence states in order to enhance the understanding of the catalyst's performance and stability.
4. To provide more comprehensive experimental details, the spectra and light intensities of the xenon lamp equipped with a 420 nm cut-off filter should be included.
5. The quantum yield is an important index to evaluate photocatalytic efficiency. The author should give the catalytic quantum yield of the test, especially the quantum yield with optimizing catalyst dosage.
6. In the in-situ FTIR experiments, the experiment details and conditions should be provided.
7. In addition, The authors proposed that the C atom of CO_2 bound with S proved by IR, and O bound with In. Why not O bind with Cu or Sn? Can authors give any reasons to exclude the possibility of O binding to Cu or Sn of CuInSnS_4 sample?
8. What is the product of oxidation half reaction in the present reaction system? The oxidation half reaction is very important for the system, and it needs to be clarified it.

Point-by-point response to the reviewers #1

Reviewer #1 (Remarks to the Author):

Zhang et al. reported CO₂ reduction activity of individual metal sulfides including In₂S₃, SnS₂, and Cu₂S and multi-metal sulfides (CuInSnS₄), and demonstrate the reduction products can be switched from CO to CH₄ by adjusting the metal to non-metal sites in metallic sulfides. Despite the detailed study of the mechanism, the inferior selectivity and poor yields severely limits such system for practical applications. As far as I know, conversions from CO₂ to CO or CO₂ to CH₄ are extensively studied, and much higher yield and selectively is reported, such as the following references [Chem. Commun., 2021,57, 8468-8471; Appl. Catal., B, 2021, 285, 119773; Adv. Funct. Mater., 2021, 31, 2010780; ACS Sustain. Chem. Eng. 2019, 7, 650-659; J. Am. Chem. Soc. 2018, 140, 14595-14598]. The novelty of this manuscript is not enough to match the high-level journal, Nature Communications, so I strongly recommend to reject it. The following issues need to be considered before publication on any journal.

Response: Thank you very much for your insightful reading of our manuscript and your helpful comments. We are sorry for the misunderstanding and confusion of the novelty of our work. Maybe, the novelty of our work is not enough to clearly highlight in the manuscript. In the field of photocatalysis research, a great variety of photocatalysts has been developed for the CO₂ conversion into various products, including CO, HCOOH, HCHO, CH₃OH, and CH₄. A pristine photocatalyst generally processes a low CO₂ conversion efficiency without addition of any scavenger due to serious radiative recombination issues. The photocatalytic performance can be enhanced by coupling with other semiconductor-based photocatalysts or precious noble metals as cocatalysts or single-atom active centers. Many studies as you referred [Chem. Commun., 2021,57, 8468-8471; Appl. Catal., B, 2021, 285, 119773; Adv. Funct. Mater., 2021, 31, 2010780; ACS Sustain. Chem. Eng. 2019, 7, 650-659; J. Am. Chem. Soc. 2018, 140, 14595-14598], showed that the modified photocatalysts can greatly improve the photocatalytic activity of CO₂ reduction, as well as the high selectivity to CH₄ products. However, achieving both high selectivity and high conversion for the photocatalytic CO₂ reduction is highly desirable and yet remains a great challenge. The identification of active center of catalysts for the adsorption and activation of CO₂ is prerequisite for the efficient CO₂ conversion and product selectivity.

Metal sulfide photocatalysts are a type of reduction semiconductor photocatalysts with narrow bandgap and negative conduction band potential, which make them have unique photocatalytic performance in solar-to-fuel conversion. Almost all studies believe that the metal center or defect site on the surface of metal sulfides is acted as main sites for CO₂ adsorption and activation¹⁻⁷. However, the non-metal sites of sulfides are very rarely considered as active center for CO₂ conversion. In our work, we find that the non-metal sulfur atom in the prepared multiple metal sulfides CuInSnS₄ octahedron nanocrystal is thermodynamically favorable to activate CO₂ and leads to a switch of main products from CO to CH₄ for the photocatalytic CO₂ reduction, as compared with the common individual metal sulfides In₂S₃, SnS₂ and Cu₂S with metal center as active sites to form CO products. Although the polymetallic sulfide CuInSnS₄ only exhibits a CH₄ yield of 6.53 μL h⁻¹, it is noteworthy that the conversion from monometallic sulfides to polymetallic sulfides results in a shift in the main products of photocatalytic CO₂ reduction from CO to CH₄. We reveal that different adsorption configurations of CO₂ on metal sulfides lead to different products in CO₂ photoreduction. CO₂ is adsorbed on electron-poor S center of CuInSnS₄ to form S-C-O-In structural unit for the further photocatalytic multi-electron processes, which contributes to the formation of CH₄ main products. Hence, the CO₂ photocatalytic conversion mechanism revealed by this research holds immense importance and can attract the intensive interesting of researchers. Through our study, we provide an insight into the role of non-metal center of photocatalyst in determining the conversion and selectivity of photocatalytic CO₂ reduction. We believe that this knowledge can contribute to the development of more efficient and selective photocatalysts for CO₂ reduction in the future.

1. From the introduction, we can see that one key concern of this manuscript is the selectivity of CO₂ reduction. As the author put forward, high production selectivity has always been the goal of CO₂ reduction. The design strategy of catalysts presented by the author is to achieve highly efficient and selective conversion of CO₂. However, according to the catalytic results provided by the authors, the multi-metal catalyst increases the variety of gas-phase products and significantly reduces the overall selectivity when compared to Cu₂S (100% CO selectivity) and SnS₂ (>90% CO selectivity). This goes against the original intention of designing high selectivity catalysts.

Response: We are very grateful for your suggestion. As you mentioned, achieving both high selectivity and high conversion for the photocatalytic CO₂ reduction is highly desirable and yet remains a great challenge in the field of photocatalysis research. It has been well understood for the photocatalytic process that the identification of active center of catalysts for the adsorption and activation of CO₂ is prerequisite for the efficient CO₂ conversion and product selectivity. Almost all studies believe that the metal center or defect site on the surface of metal sulfide photocatalysts is acted as main sites for CO₂ adsorption and activation¹⁻⁷. However, the non-metal sites of sulfide are very rarely considered as active center for CO₂ conversion. In our work, we find that the non-metal sulfur atom in our prepared multiple metal sulfides CuInSnS₄ octahedron nanocrystal is thermodynamically favorable to activate CO₂ and leads to a switch of main products from CO to CH₄, as compared with the common individual metal sulfides In₂S₃, SnS₂ and Cu₂S with metal center as active sites to form CO products. We reveal that CO₂ is adsorbed on electron-poor S center of CuInSnS₄ to form S-C-O-In structural unit for the further photocatalytic multi-electron processes into CH₄. This work provides an insight into the role of non-metal center of photocatalyst in determining the conversion and selectivity of photocatalytic CO₂ reduction.

Indeed, Cu₂S and SnS₂ have a high selectivity for CO evolution, but the yield of CO is low. Moreover, CH₄ product is difficult to form over the pristine Cu₂S and SnS₂ photocatalysts. The pristine CuInSnS₄ only exhibits a yield of CH₄ evolution of 6.53 μL h⁻¹ (corresponding to 5.83 μmol h⁻¹ g⁻¹) with a selectivity of 67.3% based on the contents of carbon-containing products. The activity and selectivity of CH₄ evolution on CuInSnS₄ can be improved by coupling with semiconductor photocatalysts or noble metals as cocatalysts. We supplementarily evaluated the photocatalytic performance of CuInSnS₄ modified with Pt, CoO_x, NiO_x, and Co(OH)₂ co-catalysts for the CO₂ reduction reaction. **Table S2** lists a comparison of the photoreduction activity of CuInSnS₄ and co-catalyst modified CuInSnS₄ photocatalysts, along with the common metal sulfide systems currently used for CO₂ photoreduction. Clearly, the modification of the co-catalyst greatly enhances the activity and selectivity of CuInSnS₄ in producing CH₄. The activity of CH₄ evolution on the modified CuInSnS₄ photocatalysts surpass the majority of the reported photocatalysts for CO₂ reduction up to now. Specifically, when Pt co-catalysts were decorated on the surface of CuInSnS₄ nano single crystals, the CO₂ photoreduction activity was significantly enhanced. The yields of CH₄ and CO were 43.25 μmol h⁻¹ g⁻¹ and 7.85 μmol h⁻¹ g⁻¹, respectively, The CH₄ selectivity was increased to

84.6%. CoO_x modified CuInSnS₄ photocatalysts yielded 33.30 μmol h⁻¹ g⁻¹ of CH₄ and 9.63 μmol h⁻¹ g⁻¹ of CO, respectively, corresponding to a CH₄ selectivity of 78.1%. NiO_x modified CuInSnS₄ photocatalysts produced 11.80 μmol h⁻¹ g⁻¹ of CH₄ and 6.85 μmol h⁻¹ g⁻¹ of CO, respectively, with a CH₄ selectivity of 63.3%. Co(OH)₂ loaded CuInSnS₄ photocatalysts had the yields of CH₄ and CO of 145.45 μmol h⁻¹ g⁻¹ and 32.32 μmol h⁻¹ g⁻¹, respectively, while the CH₄ selectivity of 81.8%. The photocatalytic CO₂ reduction performance of CuInSnS₄ was further improved by modifying with oxidation Co(OH)₂ and reduction Pt double co-catalysts. The Co(OH)₂/CuInSnS₄/Pt photocatalyst yielded 195.60 μmol h⁻¹ g⁻¹ of CH₄ and 22.00 μmol h⁻¹ g⁻¹ of CO, with a CH₄ selectivity of 89.9%. Thus, the modification of CuInSnS₄ with various co-catalysts to enhance the separation of the photogenerated charge carrier can improve both the activity and selectivity of the CH₄ products. We believe that CuInSnS₄ nano single crystal photocatalysts through further optimization design of the different contents and types of cocatalyst modification can be more efficient and selective for CO₂ reduction in the future.

Table S2 Comparing the photocatalytic CO₂ reduction performance of CuInSnS₄, modified CuInSnS₄, and common photocatalysts.

Catalyst	Generation rate of CH ₄ (μmol h ⁻¹ g ⁻¹)	Generation rate of CO (μmol h ⁻¹ g ⁻¹)	Selectivity of CH ₄	Ref.
CuInSnS ₄	5.83	2.40	67.3%	This work
Pt/CuInSnS ₄	43.25	7.85	84.6%	This work
CoO _x /CuInSnS ₄	33.30	9.36	78.1%	This work
NiO _x /CuInSnS ₄	11.80	6.85	63.3%	This work
Co(OH) ₂ /CuInSnS ₄	145.45	32.32	81.8%	This work
Co(OH) ₂ /CuInSnS ₄ /Pt	195.60	22.00	89.9%	This work
Mn(II)-metalated porphyrin	~53	~21	71.2%	8

CuS@ZnIn ₂ S ₄ /C ₆₀	43.6	6.4	87.2%	9
Pt@h-BN	9.24	0	100%	10
SnS ₂ /TiO ₂	23	2.5	90.2%	11
SiC@MoS ₂	14.42	0	100%	12
V _o -Zn-CoO	17.10	9.70	63.8%	13
In ₄ SnS ₈	23.88	20.96	57.0%	14
V _S -CuIn ₅ S ₈ single-unit-cell layers	8.70	0	100%	15
Cu ₂ O-111-Cu ⁰	78.40	14.5	97%	16
H-TiO _{2-x} (200)	16.20	4.2	79%	17
Ag ₄ /Cu ₂ O@rGO	82.6	4.0	95.4%	18
Cs ₂ SnI ₆ /SnS ₂	~6.09	0	~100%	19
Pt-Cu ₂ O/TiO ₂	1.42	0.05	96.6%	20
N-RGO/CdS	0.33	2.59	11.3%	21
Au _{SA} /Cd _{1-x} S	11.3	32.2	22.0%	22
Au _{SA} /CdS _{1-x}	0.40	3.70	9.3%	22
Ni-doped CoS ₂ nanosheets	101.8	37.5	~73.10%	23
Pd ₇ Cu ₁ -TiO ₂	19.60	1.90	95.9%	24
Ag ₂ Pd ₁ /TiO ₂	79.0	0	100%	25

2. Various work on the role of non-metals as catalytic sites for CO₂ reduction has been reported, and they have achieved efficient/highly selective conversion of CO₂. The citation of relevant literature is severely insufficient.

Response: Thank you very much for your helpful suggestion. In the revised manuscript, we have cited the relevant literatures and reviews of various metal free photocatalysts reported for CO₂ reduction, typically such as covalent organic frameworks (COFs)²⁶, Graphitic carbon nitride (g-C₃N₄)²⁷, boron nitride (BN)²⁸, elemental phosphorus (P)²⁹, and silicon carbide (SiC)³⁰. The followings are some detail examples of metal free photocatalysts for CO₂ reduction. Biswas et al. reported a study on 2D COF (TTA-Tz) composed of 1,3,5-tris-(4-aminophenyl)triazine (TTA) and 4,4'-(thiazolo[5,4-d]thiazole-2, 5-diyl)dibenzaldehyde (Tz) for photocatalytic CO₂ reduction³¹. The 2D-layered COF showed the visible-light-driven photoreduction of CO₂ to CO (yield rate = 82 μmol h⁻¹ g⁻¹, and selectivity >99%) in aqueous medium without an external sacrificial electron donor. The imine nitrogen center in 2D COF was capable of binding CO₂, to form intermediate [TTA-Tz⁻-CO₂]⁻. Xu et al. combined g-C₃N₄ nanosheets with metal-organic frameworks decorated with B-H bonding for CO₂ activation and photoreduction³². It was observed that the activation of CO₂ molecules was attributed to a high concentration of localized electrons in the near-surface region and B-H bonding sites that can trap oxygen atoms from CO₂ molecules. Consequently, the B-H bonding site functioned as an active site for CO₂ adsorption. Subsequently, upon receiving photoexcited electrons from g-C₃N₄, the B-H groups were readily triggered for CO₂ reduction due to the high localized electron concentration near the surface region. Chen et al. designed rare earth La single atoms on carbon nitride, employing La-N charge-transfer bridges as active centers for photocatalytic CO₂ reactions³³. The La-N charge bridge was identified as the crucial active center responsible for CO₂ activation, rapid formation of COOH*, and subsequent CO desorption. The electronic structure of the La-N bridge enabled a high CO yield of 92 μmol g⁻¹ h⁻¹ and a CO selectivity of 80.3%. Feng et al. incorporated nonpolar carbon quantum dots (CQDs) onto g-C₃N₄, creating a metal-free heterojunction photocatalyst (CQDs/g-C₃N₄) for the photocatalytic CO₂ reduction into CH₄³⁴. This modification resulted in a reduced band gap and the improvement in light absorption and the separation of photogenerated carriers and CO₂ adsorption. Yuan et al. developed a composite photocatalyst by coupling red phosphor (r-P) and graphitic carbonitride (g-C₃N₄), leading to a significant enhancement in the photocatalytic activity of CO₂ conversion to

valuable hydrocarbon fuels, particularly CH₄³⁵. This enhancement was attributed to the efficient separation of photogenerated electrons and holes on the r-P/g-C₃N₄ heterojunction. Cao et al. prepared ultrathin oxygen-modified h-BN (O/BN) nanosheets³⁶. The B–O bond serves as the adsorption site for CO₂, enabling the chemical capture of CO₂ and formation of an O–B–O bond. Consequently, the activation energy of CO₂ conversion was significantly reduced, resulting in excellent CO₂ photoreduction performance. Qu et al. demonstrated through theoretical calculations that the original boron nitride nanosheets or nanotubes were not effective in adsorbing and activating CO₂³⁷. However, boron-rich boron nitride nanosheets or nanotubes can effectively adsorb and activate CO₂, leading to the conversion of CO₂ into CH₄. The boron atoms on the surface of boron-rich boron nitride nanosheets or nanotubes served as chemical adsorption sites for CO₂. Hu et al. discovered that amorphous red P can selectively photoreduce CO₂ to CO³⁸. Theoretical calculations demonstrated that CO₂ molecules were adsorbed on the surface of the fibrous red P catalyst for the activation. Wang et al. designed and prepared a metal-free open β-SiC hollow sphere photocatalyst³⁹. The prepared hollow sphere SiC with unique electronic structure, hollow morphology, and high BET surface area, contributed to its high activity in converting CO₂ primarily into CH₄ products in the photocatalytic reduction of CO₂ with pure H₂O. Weng et al. achieved the one-step, template-free synthesis of silicon carbide nanowire/carbon (SiC-NW/C) composites⁴⁰. The SiC-NW/C nanostructure facilitated efficient CO₂ adsorption and rapid separation and transfer of carriers. The SiC-NW/C composite had excellent photoreduction activity of CO₂ to CO without the need for additional cocatalysts or sacrificial agents. These metal free photocatalysts have non-metallic sites as the adsorption and activation sites of CO₂ molecules and thus photocatalytic CO₂ reduction. On the contrary, for the metal oxide or sulfide photocatalysts, the metal centers or defect sites on the surface are always considered to be main sites for CO₂ adsorption and activation, but the non-metal sites are very rarely considered as active center for CO₂ conversion.

3.EDS is not a reliable method for studying element content. It is recommended that the author implement ICP-MS to determine the content and proportion of metals.

Response: Thank you for your valuable suggestions. The results of the ICP-MS tests with three times for the elemental content of the CuInSnS₄ sample were presented in **Table S1**. The atomic

ratio of Cu: In, and Sn was calculated from the mean of three measurements to be 1.06:1.00:1.00.

The detected ratio of metal atoms closely matched the stoichiometric value of CuInSnS₄.

Table S1 ICP-MS test of each element of CuInSnS₄ sample.

Sample	Quality m ₀ (g)	Elements of the test	Sample element content (W)	Sample element molar (mmol)
CuInSnS ₄	0.0192	Cu	9.75%	0.0295
		Cu	9.83%	0.0297
		Cu	9.70%	0.0293
		Sn	17.20%	0.0278
		Sn	17.15%	0.0277
		Sn	17.20%	0.0278
		In	16.71%	0.0279
		In	16.77%	0.0280
		In	16.86%	0.0282

4. For the control experiments, the production of CO was still observed under N₂ atmosphere. In the isotope labelling experiments, the authors only give the GC-MS spectra of the CH₄ product, without mentioning CO. It is recommended that the CO isotope labelling experiments be monitored and the relevant GC-MS spectra be provided.

Response: Thank you for your valuable feedback regarding the control experiments and isotope labeling experiments in our manuscript. For the control experiment, small CO products was still observed under N₂ atmosphere. This could be attributed to the interference of a small amount of CO from external sources. The out-system contamination maybe arise from the surrounding air, reactors, and equipment^{41,42}, as we can see that small CO products was also detected without CuInSnS₄ photocatalysts in the control experiment.

The ¹³CO₂ isotope experiments was provided to further confirm that the generated CO was indeed a product of CO₂ photoreduction. When the reaction was conducted in a ¹³CO₂ atmosphere, the presence of ¹³CO with a m/z = 29 was detected, as shown in **Supplementary Fig. 17a**. It is noteworthy that the mass spectrum peak at m/z = 28 corresponded to the N₂ molecule, as evident

from the distinct N_2 peaks in chromatogram. The presence of N_2 originates from air interference. As comparison, the mass spectrum peak of ^{13}CO was weak due to the low activity of $CuInSnS_4$ for CO evolution. Additionally, the $^{12}CO_2$ experiment can further evidence that the generated CO resulted from the photoreduction of CO_2 molecules. When the reaction was conducted in a $^{12}CO_2$ atmosphere, only ^{12}CO and N_2 with a $m/z = 28$ were detected. No peaks at $m/z = 29$ could be detected, as shown in **Supplementary Fig. 17b**.

Supplementary Fig. 17 (a) GC-MS spectra of ^{13}CO generated from $^{13}CO_2$, (b) GC-MS spectra of ^{12}CO generated from $^{12}CO_2$.

5. The author is suggested to pay attention to the details of the article, for example in line 64 " CO_2 ".

Response: We are sorry for any clerical errors and formatting issues in the manuscript, which have now been corrected. Please refer to the revised manuscript for the updated version. The specific revised content is as followings:

"This leads to the easy cleaving of metal-C bonds during the reaction process, hindering the deep reduction of CO_2 into hydrocarbons"

"This is attributed to the difference in the In coordinated environment between $CuInSnS_4$ and In_2S_3 because the partial In atom in In_2S_3 exists in the state of $[InS_4]$ tetrahedron"

"No macroscopic infrared absorption peaks for intermediates are found on Cu_2S or SnS_2 , even under light irradiation, possibly due to their weak chemical interaction with CO_2 (Supplementary Fig. 21a and Supplementary Fig. 21b)."

"This benefits the continuous reduction of CO_2 molecules into $COOH^$, HCO^* , H_3CO^* intermediates, and ultimately into CH_4 on the catalyst surface through the assistance of photo-*

generated electrons and protons.”

“The cutoff energy of the basis function was 420 eV. For the CuSnInS_4 (1 1 1) crystal plane, a $2 \times 2 \times 1$ super cell with a four-layer slab was constructed, and only the top two layers were allowed to relax.”

References

- [1] Li, X., Sun, Y., Xu, J., Shao, Y., Wu, J., Xu, X., Pan, Y., Ju, H., Zhu, J., & Xie, Y. Selective visible-light-driven photocatalytic CO_2 reduction to CH_4 mediated by atomically thin CuIn_5S_8 layers. *Nat. Energy* **4**, 690–699 (2019). <https://doi.org/10.1038/s41560-019-0431-1>
- [2] Wang, J., Bo, T., Shao, B., Zhang, Y., Jia, L., Tan, X., Zhou, W., & Yu, T. Effect of S vacancy in Cu_3SnS_4 on high selectivity and activity of photocatalytic CO_2 reduction. *Appl. Catal., B* **297**, 120498 (2021). <https://doi.org/10.1016/j.apcatb.2021.120498>
- [3] Si, S., Shou, H., Mao, Y., Bao, X., Zhai, G., Song, K., Wang, Z., Wang, P., Liu, Y., Zheng, Z., Dai, Y., Song, L., Huang, B. & Cheng, H. Low-coordination single Au atoms on ultrathin ZnIn_2S_4 nanosheets for selective photocatalytic CO_2 reduction towards CH_4 . *Angew. Chem. Int. Ed.* **61**, e2022030 (2022). <https://doi.org/10.1002/anie.202209446>
- [4] Cao, Y., Guo, L., Dan, M., Doronkin, D. E., Han, C., Rao, Z., Liu, Y., Meng, J., Huang, Z., Zheng, K., Chen, P., Dong, F., & Zhou, Y. Modulating electron density of vacancy site by single Au atom for effective CO_2 photoreduction. *Nat. Commun.* **12**, 1675 (2021). <https://doi.org/10.1038/s41467-021-21925-7>
- [5] Sharma, N., Das, T., Kumar, S., Bhosale, R., Kabir, M., & Ogale, S. Photocatalytic activation and reduction of CO_2 to CH_4 over single phase nano Cu_3SnS_4 : a combined experimental and theoretical study. *ACS Appl. Energy Mater.* **2**, 5677–5685 (2019). <https://doi.org/10.1021/acsaem.9b00813>
- [6] Wu, C. Y., Lee, C. J., Yu, Y. H., Tsao, H. W., Su, Y. H., Kaun, C. C., Chen, J. S., & Wu, J. J. Efficacious CO_2 photoconversion to C_2 and C_3 hydrocarbons on upright SnS-SnS_2 heterojunction nanosheet frameworks. *ACS Appl. Mater. Interfaces* **13**, 4984–4992 (2021). <https://doi.org/10.1021/acsaami.0c18420>
- [7] Tan, C. L., Qi, M. Y., Tang, Z. R., & Xu, Y. J. Isolated single-atom cobalt in the ZnIn_2S_4 monolayer with exposed Zn sites for CO_2 photofixation. *ACS Catal.* **13**, 8317–8329 (2023).

<https://doi.org/10.1021/acscatal.3c00992>

[8] Qin, J. H., Xu, P., Huang, Y. D., Xiao, L. Y., Lu, W., Yang, X. G., Ma, L. F., & Zang, S. Q. High loading of Mn(II)-metalated porphyrin in a MOF for photocatalytic CO₂ reduction in gas–solid conditions. *Chem. Commun.* **57**, 8468–8471 (2021). <https://doi.org/10.1039/D1CC02847B>

[9] Ding, Y., Chen, Y., Guan, Z., Zhao, Y., Lin, J., Jiao, Y., & Tian, G. Hierarchical CuS@ZnIn₂S₄ hollow double-shelled p–n heterojunction octahedra decorated with fullerene C₆₀ for remarkable selectivity and activity of CO₂ photoreduction into CH₄. *ACS Appl. Mater. Interfaces* **14**, 7888–7899 (2022). <https://doi.org/10.1021/acscami.1c20980>

[10] Bi, W., Hu, Y., Jiang, H., Zhang, L., & Li, C. Revealing the sudden alternation in Pt@h-BN nanoreactors for nearly 100% CO₂-to-CH₄ photoreduction. *Adv. Funct. Mater.* **31**, 2010780 (2021). <https://doi.org/10.1002/adfm.202010780>

[11] She, H., Zhou, H., Li, L., Zhao, Z., Jiang, M., Huang, J., Wang, L., & Wang, Q. Construction of a two-dimensional composite derived from TiO₂ and SnS₂ for enhanced photocatalytic reduction of CO₂ into CH₄. *ACS Sustainable Chem. Eng.* **7**, 650–659 (2019). <https://doi.org/10.1021/acssuschemeng.8b04250>

[12] Wang, Y., Zhang, Z., Zhang, L., Luo, Z., Shen, J., Lin, H., Long, J., Wu, J. C. S., Fu, X., Wang, X., & Li, C. Visible-light driven overall conversion of CO₂ and H₂O to CH₄ and O₂ on 3D-SiC@2D-MoS₂ heterostructure. *J. Am. Chem. Soc.* **140**, 14595–14598 (2018). <https://doi.org/10.1021/jacs.8b09344>

[13] Chen, K., Jiang, T., Liu, T., Yu, J., Zhou, S., Ali, A., Wang, S., Liu, Y., Zhu, L., & Xu, X. Zn dopants synergistic oxygen vacancy boosts ultrathin CoO layer for CO₂ photoreduction. *Adv. Funct. Mater.* **32**, 2109336 (2022). <https://doi.org/10.1002/adfm.202109336>

[14] Chai, Y., Chen, Y., Shen, J., Ni, M., Wang, B., Li, D., Zhang, Z., & Wang, X. Distortion of the coordination structure and high symmetry of the crystal structure in In₄SnS₈ microflowers for enhancing visible-light photocatalytic CO₂ reduction. *ACS Catal.* **11**, 11029–11039 (2021). <https://doi.org/10.1021/acscatal.1c02937>

[15] Li, X., Sun, Y., Xu, J., Shao, Y., Wu, J., Xu, X., Pan, Y., Ju, H., Zhu, J., & Xie, Y. Selective visible-light-driven photocatalytic CO₂ reduction to CH₄ mediated by atomically thin CuIn₅S₈ layers. *Nat. Energy* **4**, 690–699 (2019). <https://doi.org/10.1038/s41560-019-0431-1>

[16] Yin, G., Huang, X., Chen, T., Zhao, W., Bi, Q., Xu, J., Han, Y., & Huang, F. Hydrogenated blue

titania for efficient solar to chemical conversions: preparation, characterization, and reaction mechanism of CO₂ reduction. *ACS Catal.* **8**, 1009–1017 (2018).

<https://doi.org/10.1021/acscatal.7b03473>

[17] Tang, Z., He, W., Wang, Y., Wei, Y., Yu, X., Xiong, J., Wang, X., Zhang, X., Zhao, Z., & Liu, J. Ternary heterojunction in rGO-coated Ag/Cu₂O catalysts for boosting selective photocatalytic CO₂ reduction into CH₄. *Appl. Catal., B* **311**, 121371 (2022).

<https://doi.org/10.1016/j.apcatb.2022.121371>

[18] Wang, X. D., Huang, Y. H., Liao, J. F., Jiang, Y., Zhou, L., Zhang, X. Y. Chen, H. Y., & Kuang, D. B. In situ construction of a Cs₂SnI₆ perovskite nanocrystal/SnS₂ nanosheet heterojunction with boosted interfacial charge transfer. *J. Am. Chem. Soc.* **141**, 13434–13441 (2019).

<https://doi.org/10.1021/jacs.9b04482>

[19] Xiong, Z., Lei, Z., Kuang, C. C., Chen, X., Gong, B., Zhao, Y., Zhang, J., Zheng, C., & Wu, J. C.S. Selective photocatalytic reduction of CO₂ into CH₄ over Pt-Cu₂O TiO₂ nanocrystals: The interaction between Pt and Cu₂O cocatalysts. *Appl. Catal., B* **202**, 695–703 (2017).

<https://doi.org/10.1016/j.apcatb.2016.10.001>

[20] Bie, C., Zhu, B., Xu, F., Zhang, L., & Yu, J. In situ grown monolayer N-doped graphene on CdS hollow spheres with seamless contact for photocatalytic CO₂ reduction. *Adv. Mater.* **31**, 1902868 (2019). <https://doi.org/10.1002/adma.201902868>

[21] Cao, Y., Guo, L., Dan, M., Doronkin, D. E., Han, C., Rao, Z., Liu, Y., Meng, J., Huang, Z., Zheng, K., Chen, P., Dong, F., & Zhou, Y. Modulating electron density of vacancy site by single Au atom for effective CO₂ photoreduction. *Nat. Commun.* **12**, 1675 (2021).

<https://doi.org/10.1038/s41467-021-21925-7>

[22] Xu, J., Ju, Z., Zhang, W., Pan, Y., Zhu, J., Mao, J., Zheng, X., Fu, Ha., Yuan, M., Chen, H., & Li, R. Efficient infrared-light-driven CO₂ reduction over ultrathin metallic Ni-doped CoS₂ Nanosheets. *Angew. Chem. Int. Ed.* **60**, 8705–8709 (2021).

<https://doi.org/10.1002/anie.202017041>

[23] Long, R., Li, Y., Liu, Y., Chen, S., Zheng, X., Gao, C., He, C., Chen, N., Qi, Z., Song, L., Jiang, J., Zhu, J., & Xiong, Y. Isolation of Cu atoms in Pd lattice: forming highly selective sites for photocatalytic conversion of CO₂ to CH₄. *J. Am. Chem. Soc.* **139**, 4486–4492 (2017).

<https://doi.org/10.1021/jacs.7b00452>

- [24] Tan, D., Zhang, J., Shi, J., Li, S., Zhang, B., Tan, X., Zhang, F., Liu, L., Shao, D., & Han, B. Photocatalytic CO₂ transformation to CH₄ by Ag/Pd bimetals supported on N-Doped TiO₂ nanosheet. *ACS Appl. Mater. Interfaces* **10**, 24516–24522 (2018). <https://doi.org/10.1021/acsami.8b06320>
- [25] Jiao, X., Chen, Z., Li, X., Sun, Y., Gao, S., Yan, W., Wang, C., Zhang, Q., Lin, Y., Luo, Y., & Xie, Y. Defect-mediated electron–hole separation in one-unit-cell ZnIn₂S₄ layers for boosted solar-driven CO₂ reduction. *J. Am. Chem. Soc.* **139**, 7586–7594 (2017). <https://doi.org/10.1021/jacs.7b02290>
- [26] Nguyen, H. L., & Alzamy, A. Covalent organic frameworks as emerging platforms for CO₂ photoreduction. *ACS Catal.* **11**, 9809–9824 (2021). <https://doi.org/10.1021/acscatal.1c02459>
- [27] Guo, R.T., Zhang, Z.R., Xia, C., Li, C.F., & Pan, W.G. Recent progress of cocatalysts loaded on carbon nitride for selective photoreduction of CO₂ to CH₄. *Nanoscale* **15**, 8548–8577 (2023). <https://doi.org/10.1039/D3NR00242J>
- [28] Laghaei, M., Ghasemian, M., Lei, W., Kong, L., & Chao, Q. A review of boron nitride-based photocatalysts for carbon dioxide reduction. *J. Mater. Chem. A*, **11**, 11925–11963 (2023). <https://doi.org/10.1039/D2TA09564E>
- [29] Zhai, R., Zhang, L., Gu, M., Zhao, X., Zhang, B., Cheng, Y., & Zhang, J. A review of phosphorus structures as CO₂ reduction photocatalysts. *Small* **19**, 2207840 (2023). <https://doi.org/10.1002/smll.202207840>
- [30] Li, C., Xu, Y., Tu, W., Chen, G., & Xu, R. Metal-free photocatalysts for various applications in energy conversion and environmental purification. *Green Chem.* **19**, 882–899 (2017). <https://doi.org/10.1039/C6GC02856J>
- [31] Biswas, S., Dey, A., Rahimi, F. A., Barman, S., & Maji, T. K. Metal-free highly stable and crystalline covalent organic nanosheet for visible-light-driven selective solar fuel production in aqueous medium. *ACS Catal.* **13**, 5926–5937 (2023) <https://doi.org/10.1021/acscatal.2c05203>
- [32] Xu, G., Zhang, H., Wei, J., Zhang, H.-X., Wu, X., Li, Y., Li, C., Zhang, J., & Ye, J. Integrating the g-C₃N₄ nanosheet with B–H bonding decorated metal–organic framework for CO₂ activation and photoreduction. *ACS Nano* **12**, 5333–5340 (2018). <https://doi.org/10.1021/acsnano.8b00110>
- [33] Chen, P., Lei, B., Dong, X., Wang, H., Sheng, J., Cui, W., Li, J., Sun, Y., Wang, Z., & Dong, F. Rare-earth single-atom La–N charge-transfer bridge on carbon nitride for highly efficient and

selective photocatalytic CO₂ reduction. *ACS Nano* **14**, 15841–15852 (2020).
<https://doi.org/10.1021/acsnano.0c07083>

[34] Feng, H., Guo, Q., Xu, Y., Chen, T., Zhou, Y., Wang, Y., Wang, M., & Shen, D. Surface nonpolarization of g-C₃N₄ by decoration with sensitized quantum dots for improved CO₂ photoreduction. *ChemSusChem* **11**, 4256–4261 (2018). <https://doi.org/10.1002/cssc.201802065>

[35] Yuan, Y. P., Cao, S. W., Liao, Y. S., Yin, L. S., & Xue, C. Red phosphor/g-C₃N₄ heterojunction with enhanced photocatalytic activities for solar fuels production. *Appl. Catal., B* **140-141**, 164-168 (2013). <https://doi.org/10.1016/j.apcatb.2013.04.006>

[36] Cao, Y., Zhang, R., Zhou, T., Jin, S., Huang, J., Ye, L., Huang, Z., Wang, F., & Zhou, Y. B–O bonds in ultrathin boron nitride nanosheets to promote photocatalytic carbon dioxide conversion. *ACS Appl. Mater. Interfaces* **12**, 9935–9943 (2020). <https://doi.org/10.1021/acsmi.9b21157>

[37] Qu, M., Qin, G., Fan, J., Du, A., & Sun, Q. Boron-rich boron nitride nanomaterials as efficient metal-free catalysts for converting CO₂ into valuable fuel. *Appl. Surf. Sci.* **555**, 149652 (2021).
<https://doi.org/10.1016/j.apsusc.2021.149652>

[38] Hu, Z., Lu, Y., Liu, M., Zhang, X., & Cai, J. Crystalline red phosphorus for selective photocatalytic reduction of CO₂ into CO. *J. Mater. Chem. A*, **9**, 338-348 (2021).
<https://doi.org/10.1039/D0TA09177D>

[39] Wang, Y., Zhang, L., Zhang, X., Zhang, Z., Tong, Y., Li, F., Wu, J. C.-S., & Wang, X. Openmouthed β-SiC hollow-sphere with highly photocatalytic activity for reduction of CO₂ with H₂O. *Appl. Catal., B* **206**, 159-167 (2017). <https://doi.org/10.1016/j.apcatb.2017.01.028>

[40] Weng, W., Wang, S., Xiao, W., & Lou, X. W. Direct conversion of rice husks to nanostructured SiC/C for CO₂ photoreduction. *Adv. Mater.* **32**, 2001560 (2020).
<https://doi.org/10.1002/adma.202001560>

[41] Zhang, Y., Yao, D., Xia, B., Jaroniec, M., Ran, J. & Qiao, S. Z., Photocatalytic CO₂ reduction: identification and elimination of false-positive results. *ACS Energy Lett.* **7**, 1611–1617 (2022)
<https://doi.org/10.1021/acsenergylett.2c00427>

[42] Williams, G., Seger, B., & Kamat, P. V. TiO₂-Graphene nanocomposites UV-assisted photocatalytic reduction of graphene oxide. *ACS Nano* **2** 1487–1491 (2008)
<https://doi.org/10.1021/nn800251f>

Point-by-point response to the reviewers #2

This manuscript investigated the effect of metallic and nonmetallic sites as CO₂ adsorption sites on the product selectivity and activity of photocatalytic CO₂ reduction. The results demonstrated that the electron-deficient central S atom on the crystal plane of multinary metal sulfide CuSnInS₄ (1 1 1) served as the CO₂ adsorption and activation site. A stable transition state [S-C-O-In] was formed to allow CH₄ to be efficiently generated through the conversion pathway of COOH*→HCOOH*→H₂CO*→H₃CO*→CH₄*. However, the asymmetric activation of CO₂ by monometallic sulfides (In₂S₃) was more likely to result in the cleavage of a single C–O bond in the CO₂ molecule. This led to the preferential photoreduction of CO₂ to CO. This work is of significant importance for the study of photocatalytic CO₂ conversion from the perspective of catalysis, including the adsorption sites of substrate molecules, adsorption configurations, energy barriers of reaction intermediates, desorption of products, and other related fields. Therefore, the manuscript can be accepted for publication after the following improvement.

Response: We appreciate for your positive evaluation of our manuscript and helpful suggestions. We have carefully addressed your comments and suggestions in our revised manuscript. We have made efforts to improve the clarity of our explanations, provide additional supporting data, and ensure the scientific rigor of our findings. We believe that these improvements have strengthened the quality and impact of our work.

Detailed recommendations are as follows:

1. The XPS tests indicated that the In 3d binding energy of CuInSnS₄ was uniformly shifted to a lower binding energy compared to In₂S₃. Moreover, the binding energy of S atoms increased in the order of Cu₂S < In₂S₃ < SnS₂ < CuInSnS₄. These results served as evidence that S atoms lacked electrons while In atoms gained electrons on the (1 1 1) crystal face of CuInSnS₄ nano single crystal. Similarly, on the In₂S₃ (0 0 1) crystal face, the S atoms were electron-rich, and the In atoms were electron-deficient. Therefore, the author should provide a further analysis and discussion of the XPS results to support their findings.

Fig. 2 Characterization of electronic structure. High-resolution XPS spectra of metal sulfides: (a) Cu 2p, (b) In 3d, (c) Sn 3d, and (d) S 2p.

Response: We appreciate the reviewer's attention to the XPS results presented in our manuscript. The XPS tests indeed revealed significant differences in the binding energy of In and S atoms between CuInSnS₄ and In₂S₃, providing important insights into the electronic properties of these materials and supporting our findings regarding the electron-rich and electron-deficient nature of S and In atoms on different crystal faces. We have carefully re-analyzed the XPS results and provided further analysis and discussion to better support our findings. Specifically, we have focused on the In 3d binding energy shift observed in CuInSnS₄ compared to In₂S₃. The uniform shift to a lower binding energy indicates a transfer of electrons from the S atoms to the In atoms, resulting in an electron-rich state for In atoms and an electron-deficient state for S atoms on the (1 1 1) crystal face of CuInSnS₄ nano single crystal (**Fig. 2b**). This electron transfer process is crucial for the CO₂ adsorption and activation on the catalyst surface, which subsequently affects the photocatalytic CO₂ conversion activity and product selectivity. Furthermore, we have discussed the increasing binding energy trend of S atoms observed in Cu₂S, In₂S₃, SnS₂, and CuInSnS₄. This trend provides additional evidence for the electron-deficient nature of S atoms on the (1 1 1) crystal face of CuInSnS₄ (**Fig. 2d**). The higher binding energy of S atoms in CuInSnS₄ compared to the

other sulfides indicates a lower electron density in the S atom, further supporting its role as a CO₂ absorption and activation site.

We have incorporated these additional analyzes and discussions into the revised manuscript, highlighting the implications of the XPS results for our findings on the electronic properties and absorption characteristics of CuInSnS₄ and In₂S₃. We believe that these modifications have strengthened the scientific basis of our study and provide a more comprehensive understanding of the role of electron-rich and electron-deficient sites in the photocatalytic CO₂ reduction process.

2.The authors claimed that the EDS analysis showed the atomic ratio of Cu, Sn, In, and S to be approximately 1:1:1:4, which was said to be very close to the stoichiometric value of CuInSnS₄ compound, thus indicating high purity of CuInSnS₄ nanocrystals. However, this description was inadequate. The authors should provide specific atomic ratios of Cu, Sn, In, and S to accurately determine the proximity to the stoichiometric value of the CuInSnS₄ compound.

Response: Thanks for your important suggestions. The results of the ICP-MS tests with three times for the elemental content of the CuInSnS₄ sample were presented in **Table S1**. The atomic ratio of Cu: In, and Sn was calculated from the mean of three measurements to be 1.06:1.00:1.00. The detected ratio of metal atoms closely matched the stoichiometric value of CuInSnS₄.

Table S1 ICP-MS test of each element of CuInSnS₄ sample.

Sample	Quality m ₀ (g)	Elements of the test	Sample element content (W)	Sample element molar (mmol)
CuInSnS ₄	0.0192	Cu	9.75%	0.0295
		Cu	9.83%	0.0297
		Cu	9.70%	0.0293
		Sn	17.20%	0.0278
		Sn	17.15%	0.0277
		Sn	17.20%	0.0278
		In	16.71%	0.0279
		In	16.77%	0.0280
		In	16.86%	0.0282

3. The addition of water resulted in a significant infrared peak of water in CuInSnS_4 nano-single crystals, as shown in Fig. 5b, whereas In_2S_3 did not exhibit an obvious water adsorption IR peak. why?

Response: Thank you for your insightful comment regarding the addition of water and the observed differences in water adsorption between CuInSnS_4 nano-single crystals and In_2S_3 . The presence of a significant infrared peak of water in CuInSnS_4 nano-single crystals, as shown in Fig. 5b, indicates the adsorption of water molecules onto the surface of the catalyst. The absence of an obvious water adsorption IR peak in In_2S_3 could be attributed to the different surface properties of the two materials in the affinity towards water adsorption. To obtain a comprehensive understanding of the underlying factors contributing to the observed variations in water adsorption, we employed contact angle measurements to evaluate the water adsorption and wetting characteristics on the catalyst surface (**Supplementary Fig. 22**). The results revealed that the contact angle of CuInSnS_4 is 20.591° , which is smaller than that of In_2S_3 (26.657°). Consequently, water molecules exhibit a higher tendency to be adsorbed on the CuInSnS_4 surface, while the adsorption on the In_2S_3 surface is comparatively weaker.

Supplementary Fig. 22 Contact angles of CuInSnS_4 and In_2S_3 .

4. Why is this reaction considered a gas-solid reaction? It appears that during the photocatalytic CO_2 reduction test, 50 μL of deionized water was injected into the quartz reactor. The authors claimed that the deionized water was evaporated by heating it with a hair dryer. However, if the water exists in the form of water vapor during the reaction, it would mean that the reaction temperature is at least 100°C or higher. In reality, the test was conducted under normal temperature

and pressure conditions. Therefore, the reaction should be classified as a gas-liquid-solid three-phase reaction.

Response: Thank you very much for your suggestions. We are sorry for the misunderstanding and confusion of the reaction system. Throughout the entire photocatalytic CO₂ reduction process, only 50 μL of water was injected into the reactor using a liquid syringe, and the reactor was heated with a hair dryer to evaporate the water into water vapor. As a result, water is present in the form of water vapor throughout the reaction. Thus, the reaction is a gas-solid reaction.

5. The authors suggested that different coordination environments of S atoms on the In₂S₃ and CuInSnS₄ surfaces, as well as the difference in charge density on the S atoms, led to different adsorption configurations of CO₂, as shown in Supplementary Fig. 22 and Table S1. However, as seen from the schematic diagram of the crystal structure, the S atom of In₂S₃ was mainly coordinated with three In atoms, regardless of whether the crystal face was exposed or not, whereas the S atom of CuInSnS₄ was mainly coordinated with one In atom, one Cu atom, and one Sn atom. Since CuInSnS₄ mainly exposed the (1 1 1) crystal plane, the plane primarily distributed S and In atoms, whereas Cu and Sn atoms were located in deeper layers, rendering them unsuitable for CO₂ adsorption sites. Therefore, the exposure of different crystal planes, as well as the different coordination environments of S atoms and differences in charge density on S atoms, were the reasons for the different adsorption configurations of CO₂, not just the different coordination environments of surface S atoms and the differences in S atom charge densities.

Response: Thank you for your insightful comments regarding the different adsorption configurations of CO₂ on the surfaces of In₂S₃ and CuInSnS₄. Upon reviewing the schematic diagram of the crystal structure, it becomes apparent that the S atom in In₂S₃ predominantly forms coordination bonds with three In atoms, regardless of the exposure of the crystal face. In contrast, the S atom in CuInSnS₄ coordinates with one In atom, one Cu atom, and one Sn atom. Since CuInSnS₄ primarily exposes the (1 1 1) crystal plane, which consists primarily of S and In atoms, while Cu and Sn atoms are situated in deeper layers, thus the Cu and Sn atoms are less conducive to CO₂ adsorption. Therefore, the different adsorption configurations of CO₂ can be attributed to a combination of factors, including the exposure of different crystal planes, the diverse coordination environments of S atoms, and the differences in charge density on S atoms. It is evident that the

coordination environments of surface S atoms and the charge densities of S atoms alone do not fully account for the observed differences in CO₂ adsorption configurations.

6. The authors explained the differences in the photocatalytic CO₂ reduction activity between monometallic sulfides and CuInSnSn₄ nano single crystals by carrier separation efficiency and transfer rate. Additionally, they attributed the difference in product selectivity to the difference in adsorption configuration. However, it was worth noting that carrier separation efficiency and mobility rate were also responsible for the difference in CO₂ photoreduction product selectivity since CH₄ (8-electron reduction product) and CO (2-electron reduction product) required different amounts of photogenerated electrons. Furthermore, the degree of activation of the catalyst for CO₂ adsorption can also contribute to the difference in CO₂ photoreduction activity since a strong adsorption and activation ability can reduce the energy barrier of the reaction and facilitate the efficient conversion of CO₂.

Response: We sincerely appreciate the insightful comments regarding our explanation of the differences in photocatalytic CO₂ reduction activity and product selectivity between monometallic sulfides and CuInSnSn₄ nano single crystals. Your observations and suggestions have prompted us to further analyze the underlying factors influencing these processes. In our initial explanation, we focused on highlighting the role of carrier separation efficiency and migration rate in the observed variations. While these factors are indeed critical, we acknowledge that carrier separation efficiency and mobility rate are also responsible for the differences in product selectivity for CO₂ photoreduction. This is because the formation of different products, such as CH₄ (8-electron reduction product) and CO (2-electron reduction product), requires varying amounts of photogenerated electrons. By considering carrier separation efficiency and mobility rate, we can better comprehend the mechanisms governing product selectivity. Furthermore, we agree that the degree of catalyst activation for CO₂ adsorption is an essential factor influencing CO₂ photoreduction activity. A catalyst's ability to adsorb and activate CO₂ molecules is crucial for facilitating their conversion into desirable products. A catalyst with a strong adsorption and activation ability can effectively lower the energy barrier of the reaction, thereby enhancing the efficiency of CO₂ conversion. By considering the degree of catalyst activation, we can gain a more comprehensive understanding of the variations in CO₂ photoreduction activity observed between

different catalysts. In light of these valuable insights, we revise our manuscript to include a more comprehensive analysis of the factors affecting CO₂ photoreduction activity and product selectivity. We emphasize the interplay between carrier separation efficiency, mobility rate, catalyst activation, and adsorption configuration to provide a more accurate and comprehensive understanding of the observed differences. By incorporating these additional considerations, we aim to enhance the scientific rigor and clarity of our manuscript. We are grateful for your constructive feedback, which has allowed us to delve deeper into the complexities of CO₂ photoreduction and the factors influencing its efficiency and selectivity.

7. Additionally, (1) It was important to note that (1 1 1) was a crystal plane but not a surface. The author has erroneously referred to (1 1 1) as the surface of CuInSnS₄ in several instances throughout manuscript. (2) Photocurrent scale was missed in Fig 4c. (3) The authors should provide details about the light source used for photocatalytic CO₂ reduction, including light intensity and spectrum.

Response: We sincerely appreciate your additional comments.

Firstly, we are sorry for the confusion caused by our erroneous reference to (1 1 1) as the surface of CuInSnS₄. We acknowledge that (1 1 1) represents a crystal plane and not the surface. We make the necessary revisions in our manuscript to correct this misunderstanding and ensure accurate terminology is used throughout.

Secondly, the missing photocurrent scale was added in Fig. 4c to enhance the clarity and interpretation of the data.

Lastly, we agree that providing details about the light source used for photocatalytic CO₂ reduction is essential for a thorough understanding of the experimental setup. We supplement the relevant information about the light source in our revised manuscript, including the light intensity and spectrum.

The specific revised content is as follows:

“Adsorption of CO₂ on the In₂S₃ (0 0 1) crystal plane leads to inconsistent changes in two C=O lengths.”

“In the case of the CuInSnS₄ (1 1 1) crystal plane, both C-O bonds are similar in length, measuring

$1.26 \pm 0.02 \text{ \AA}$.”

“For the CuSnInS_4 (1 1 1) crystal plane, a $2 \times 2 \times 1$ super cell with a four-layer slab was constructed, and only the top two layers were allowed to relax.”

As shown in Fig. 4c, the photocurrent scale is added.

Fig. 4 Characterization of energy bands and optoelectronic properties. (c) Photocurrent response of the as-prepared samples.

Supplementary Fig. 11 Xenon light source spectrum and light intensity.

Point-by-point response to the reviewers #3

Reviewer #3 (Remarks to the Author):

In this manuscript, the authors reported that a variety of metal sulfide photocatalysts (In_2S_3 , SnS_2 , Cu_2S , CuInSnS_4) were synthesized using a simple hydrothermal method, and their application in the photocatalytic conversion of CO_2 with H_2O was investigated. The photocatalytic performance tests revealed that, compared to the single metal sulfides, the polymetallic sulfide CuInSnS_4 exhibited higher activity for CO_2 conversion and also showed a change in product selectivity from CO to CH_4 . Through systematic characterization and testing, the authors elucidated the CO_2 conversion on the S atom sites of multimetal sulfide CuInSnS_4 different from the metal sites of common single metal sulfides. Overall, the manuscript was well-prepared and well-characterized, and it will be of interest to researchers in the field of photocatalysis research. Therefore, I recommend publishing this work with appropriate modifications. Here are my comments and suggestions.

Response: Thank you for your positive feedback on our manuscript. We appreciate your time and effort in reviewing our work and providing valuable comments and suggestions. We have carefully considered your feedback and have made the necessary modifications to improve the clarity and quality of our manuscript.

1. Please provide the standard XRD patterns of the single metal sulfide samples, including In_2S_3 , SnS_2 , and Cu_2S .

Response: We appreciate your suggestion to include this important information in our manuscript to provide a more comprehensive characterization of the materials. In our revised manuscript, we include the standard XRD patterns of In_2S_3 , SnS_2 , and Cu_2S . These XRD patterns serve as a reference for the crystal structure analysis of the individual sulfide samples and provide further support for our characterization results. By including these XRD patterns, we aim to enhance the clarity and completeness of our study.

Supplementary Fig. 2 XRD patterns of In_2S_3 , SnS_2 , and Cu_2S .

2. The polymetallic sulfide CuInSnS_4 exhibits octahedral nanoparticles with exposed (1 1 1) facets. However, it is not clear to the morphology characteristics of monometallic sulfides (In_2S_3 , SnS_2 , and Cu_2S). It is recommended that the authors supplement the SEM or TEM images of monometallic sulfides.

Response: Thank you for your valuable feedback regarding the morphology characteristics of monometallic sulfides (In_2S_3 , SnS_2 , and Cu_2S) in our study. To address this concern, we include SEM images of the monometallic sulfides in the revised manuscript.

Supplementary Fig. 7 shows the SEM images of In_2S_3 , Cu_2S , and SnS_2 samples. In_2S_3 exhibits a morphology of microspheres self-assembled from nanosheets. Cu_2S demonstrates the basic shape of nanoparticles, while SnS_2 displays the morphology of ultrathin nanosheets.

Supplementary Fig. 7 SEM images of (a) In_2S_3 , (b) Cu_2S , and (c) SnS_2 .

3. The authors should provide a more comprehensive analysis and discussion of the XPS results, including the determination of the chemical valence states of each element in the CuInSnS_4 sample before and after the reaction. This detailed characterization will allow for a better understanding of the CuInSnS_4 sample. Furthermore, by examining the change in the valence state of the catalyst

before and after the reaction, the stability of the catalyst can be assessed. Hence, it is recommended to include a detailed analysis of the valence states in order to enhance the understanding of the catalyst's performance and stability.

Response: We appreciate the reviewer's suggestion regarding the analysis of the chemical valence states of each element in the CuInSnS_4 sample before and after the reaction. Understanding the valence states is indeed crucial for gaining insights into the catalyst's performance and stability. In the revised manuscript, we have included a more comprehensive analysis and discussion of the XPS results, specifically focusing on the determination of the chemical valence states of Cu, In, Sn, and S in the CuInSnS_4 sample. By analyzing the XPS spectra and employing established methods for valence state determination, we have identified the chemical valence states of each element. We have provided detailed information on the valence states of Cu, In, Sn, and S before and after the photocatalytic CO_2 reduction reaction. This analysis allows for a better understanding of the electronic state changes occurring in the catalyst during the reaction process.

Fig. 2 Characterization of electronic structure. High-resolution XPS spectra of metal sulfides: (a) Cu 2p, (b) In 3d, (c) Sn 3d, and (d) S 2p.

X-ray photoelectron spectroscopy (XPS) was used to compare the electronic states of the

obtained sample. The Cu 2p_{3/2} and Cu 2p_{1/2} binding energies of CuInSnS₄ sample are measured to be 932.07 eV and 951.90 eV, respectively (**Fig. 2a**). This demonstrates that the valence state of Cu is +1 in the CuInSnS₄ sample, which is also confirmed by the Cu LMM spectra (**Supplementary Fig. 8**). Notably, the Cu 2p binding energies of the CuInSnS₄ sample are identical to those of Cu₂S. The binding energies of In3d_{5/2} and In3d_{3/2} in the CuInSnS₄ sample are 444.63 eV and 452.18 eV, respectively. These values indicate that the valence state of In in the CuInSnS₄ sample is +3. Compared with In₂S₃, the In 3d binding energy of CuInSnS₄ uniformly shifts toward the lower binding energy, as shown in Fig. 2b. This shift can be attributed to the difference in the In coordinated environment between CuInSnS₄ and In₂S₃, as the partial In atoms in In₂S₃ exist in the state of [InS₄] tetrahedron. In the CuInSnS₄ sample, the Sn 3d_{5/2} and Sn 3d_{3/2} doublets are centered at 486.30 eV and 494.70 eV, respectively, indicating the presence of Sn⁴⁺ valence state. Notably, the binding energy of Sn in the CuInSnS₄ sample is slightly lower than that in the parent SnS₂ (**Fig. 2c**). The possible reason for this difference is that the Sn atoms are in different crystal structures. Furthermore, the binding energies of S 2p_{3/2} and S 2p_{1/2} in the CuInSnS₄ sample are measured to be 161.45 eV and 162.70 eV, respectively, which correspond to the S²⁻ valence state. In the S 2p XPS spectra, the binding energies of S atoms increase in the order of Cu₂S < In₂S₃ < SnS₂ < CuInSnS₄, as shown in **Fig. 2d**. S atoms in CuInSnS₄ have the highest binding energy. This can be interpreted by the fact that the average bond length (0.253 nm) between sulfur and metal atom in CuInSnS₄ is slightly larger than in monometallic sulfide.

The XPS spectra of the CuInSnS₄ photocatalyst before and after the reaction are presented in **Supplementary Fig. 15**. After the reaction, there is a slight increase in the binding energies of In, Cu, and S in the samples, while the binding energy of Sn remains relatively unchanged. Specifically, after the reaction, the binding energies of In 3d_{5/2} and In 3d_{3/2} in the CuInSnS₄ photocatalyst were measured to be 444.78 and 452.33 eV, respectively, indicating the presence of In³⁺ (**Supplementary Fig. 15a**). The Cu 2p_{3/2} and Cu 2p_{1/2} binding energies of CuInSnS₄ sample, measured after the reaction, were measured to be 932.52 and 952.43 eV, respectively. This demonstrates that the valence state of Cu is +1 in the CuInSnS₄ sample (**Supplementary Fig. 15b**). Similarly, in the reacted CuInSnS₄ sample, the Sn 3d_{5/2} and Sn 3d_{3/2} doublets were centered at 486.42 and 494.84 eV, respectively, confirming the persistence of the Sn⁴⁺ valence state

(Supplementary Fig. 15c). The binding energies of S 2p_{3/2} and S 2p_{1/2} in the CuInSnS₄ sample, measured after the reaction, were determined to be 161.66 and 162.84 eV, respectively, corresponding to the S²⁻ valence state. Additionally, a stable S-C-O-In adsorption configuration is created on the (1 1 1) crystal plane of the CuInSnS₄ sample due to the adsorption and activation of CO₂ molecules. This adsorption configuration enables the transfer of electrons from the (1 1 1) facet of the CuInSnS₄ sample to CO₂ molecules, leading to the activation of CO₂ molecules. As a result, the charge density of the indium atoms decreases, and the binding energy increases. S atoms transfer electrons to carbon atoms of CO₂, while Cu atoms can transfer electrons to sulfur atoms. Therefore, the degree of increase in the binding energy of S atoms is lower than that of Cu atoms. It is worth noting that the surface lattice S²⁻ in the metal sulfide of the photocatalyst can be oxidized to SO₃²⁻ or SO₄²⁻ by photogenerated holes if the photocatalyst undergoes photocorrosion, resulting in changes in the surface structures of the metal sulfide in the composite catalysts^{1,2}. However, the XPS spectrum of S element after the reaction shows only a doublet attributed to lattice S²⁻ and a very faint XPS peak of SO₃²⁻ species, as shown in **Supplementary Fig. 15d**. For the reacted catalyst, the XPS peaks with binding energies of 161.66 and 162.84 eV are assigned to S 2p_{3/2} and S 2p_{1/2}, while the new peaks with binding energies in the range of 168.26~170.26 eV are assigned to the XPS peaks of SO₃²⁻ species.^{3,4} The appearance of SO₃²⁻ species indicates that photocorrosion occurs in the photocatalyst during the long-term reaction process.

Supplementary Fig. 15 XPS spectra of CuInSnS₄ before and after the reaction, (a) In 3d, (b) Cu 2p, (c) Sn 3d, and (d) S 2p.

4. To provide more comprehensive experimental details, the spectra and light intensities of the xenon lamp equipped with a 420 nm cut-off filter should be included.

Response: Thank you for your suggestion. **Supplementary Fig. 11** shows the supplemented spectra for a xenon lamp equipped with a 420 nm cut-off filter. The light intensity of the lamp is 474 mW cm⁻².

Supplementary Fig. 11 Xenon light source spectrum and light intensity.

5. The quantum yield is an important index to evaluate photocatalytic efficiency. The author should give the catalytic quantum yield of the test, especially the quantum yield with optimizing catalyst dosage.

Response: Thank you for your valuable feedback regarding the quantum yield of our photocatalytic test. We acknowledge the importance of this parameter in evaluating photocatalytic efficiency. Due to the low yields of CH₄ and CO on the pristine CuInSnS₄, we used the Co(OH)₂ and Pt dual cocatalyst modified photocatalyst Co(OH)₂/CuInSnS₄/Pt for the calculation of the apparent quantum yield (AQY) for the CH₄ and CO yield under monochromatic light at 400 nm. When the Co(OH)₂/CuInSnS₄/Pt composite sample were irradiated with 400 nm monochromatic light, the yields of CH₄ and CO were 1.165 μmol h⁻¹ (corresponding to 23.30 μmol h⁻¹ g⁻¹) and 0.375 μmol h⁻¹ (corresponding to 7.5 μmol h⁻¹ g⁻¹), respectively. The spectrum and light intensity of 400 nm monochromatic light were measured twice in parallel, as shown in **Supplementary Fig. 18**. The light intensity values for the two measurements were 79.5 mW cm⁻² and 75.8 mW cm⁻², respectively. Therefore, the average value of the two measured values was taken as the light intensity value of the 400 nm monochromatic light.

Supplementary Fig. 18 Spectrum and intensity of 400 nm monochromatic light. (a) First test, and (b) Second test.

The calculation formula for apparent quantum efficiency was as followings. The corresponding apparent quantum efficiencies for CH₄ and CO yields over Co(OH)₂/CuInSnS₄/Pt photocatalyst were 0.16% and 0.01% at 400 nm, respectively.

$$N_{\text{photos}} = \frac{I \times S \times T}{h \frac{c}{\lambda}} = \frac{77.65 \times 1.4 \times 1.4 \times 3.14 \times 10^{-3} \times 3600}{6.626 \times 10^{-34} \times \frac{3 \times 10^8}{400 \times 10^{-9}}} = 3.46 \times 10^{21}$$

$$N_{\text{CH}_4} = 1.165 \times 10^{-6} \times 6.02 \times 10^{23} = 7.01 \times 10^{17}$$

$$N_{\text{CO}} = 0.375 \times 10^{-6} \times 6.02 \times 10^{23} = 2.26 \times 10^{17}$$

$$400\text{nm } \text{AQY}_{\text{CH}_4} = \frac{N_{\text{CH}_4} \times 8}{N_{\text{photos}}} = \frac{7.01 \times 10^{17} \times 8}{3.46 \times 10^{21}} \times 100\% = 0.16\%$$

$$400\text{nm } \text{AQY}_{\text{CO}} = \frac{N_{\text{CO}} \times 2}{N_{\text{photos}}} = \frac{2.26 \times 10^{17} \times 2}{3.46 \times 10^{21}} \times 100\% = 0.01\%$$

6. In the in-situ FTIR experiments, the experiment details and conditions should be provided.

Response: Thank you for your valuable feedback. In the revised manuscript, we include a comprehensive description of the experiment details and conditions for the in-situ FTIR measurements. This will help provide a better understanding of the experimental setup and facilitate the interpretation of the obtained results. We appreciate your suggestion and will make the necessary additions to improve the clarity and completeness of the manuscript.

In situ FTIR experiments were tested as follows: In situ infrared spectra measurements were performed using a Fourier transform infrared (FTIR) spectrometer (Nicolet iS50 FTIR Spectrometer) equipped with a mercury cadmium telluride detector (**Supplementary Fig. 30**). In situ infrared spectra were recorded by averaging 32 scans at a resolution of 4 cm^{-1} . To initiate the experiment, the catalyst was placed in a 250 mL quartz tube and compacted into a film. The tube was then subjected to vacuum treatment for 60 min. Subsequently, high-purity CO_2 gas was introduced, and the quartz tube was sealed. A liquid sampler was used to inject 60 μL of deionized water into the sealed quartz tube. The tube was heated with a hot blower to vaporize the deionized water. The quartz tube was positioned in the Fourier transform infrared (FTIR) spectrometer, ensuring that the incident light of the infrared spectrometer was perpendicular to the sample surface. In addition, a xenon lamp visible light source was introduced to directly illuminate the sample surface. Infrared spectra were recorded after pretreating the catalyst in vacuum, introducing CO_2 gas, and vaporizing deionized water, respectively. After the introduction of light, infrared spectra were recorded every 5 min.

Supplementary Fig. 30 In situ infrared testing device.

7. In addition, the authors proposed that the C atom of CO_2 bound with S proved by IR, and O bound with In. Why not O bind with Cu or Sn? Can authors give any reasons to exclude the possibility of O binding to Cu or Sn of CuInSnS_4 sample?

Response: We appreciate your question regarding the binding of oxygen (O) with copper (Cu) or tin (Sn) in the CuInSnS_4 sample. In our study, based on the in-situ infrared (IR) spectra analysis, we observed the binding of C with sulfur (S) rather than In, Cu or Sn. When the carbon (C) atom

of CO₂ binds to the sulfur (S) atom on the (1 1 1) crystal plane of the CuInSnS₄ sample, the oxygen (O) atom can only form a bond with the indium (In) atom and not with the copper (Cu) or tin (Sn) atoms. This is due to the distribution of atoms on the (1 1 1) crystal plane of the CuInSnS₄ sample. The (1 1 1) crystal plane primarily consists of S atoms, In atoms, and some Sn atoms, while the Cu atoms are predominantly located in the deeper layers, as depicted in the figure. Consequently, when the C atom of the CO₂ molecule bonds with the S atom on the (1 1 1) crystal plane, the O atom cannot form a bond with the Cu atom situated deeper within the crystal plane. Simultaneously, we provide a computational model for Cu atoms as potential CO₂ adsorption sites, as depicted in **Supplementary Fig. 25g-Supplementary Fig. 25i**. However, upon optimization, it was discovered that the aforementioned adsorption models were all deemed invalid configurations. Consequently, Cu atoms were excluded as reactive sites for CO₂ adsorption activation. Furthermore, we also examined the bonded structures of O atoms and Sn atoms individually, as shown in **Supplementary Fig. 25a-Supplementary Fig. 25f**. Our findings indicate that the adsorption configuration involving O atoms bonded to Sn atoms is ineffective compared to O atoms bonded to In atoms. In other words, when C atoms are adsorbed onto S atoms and O atoms are adsorbed onto Sn atoms, this particular adsorption configuration is unstable. Hence, we have excluded the possibility of the O atom in the CO₂ molecule bonding to the Sn atom.

Supplementary Fig. 25 Established structural models of (a-f) Sn atoms and (g-i) Cu atoms as CO₂ adsorption sites, respectively.

8. What is the product of oxidation half reaction in the present reaction system? The oxidation half reaction is very important for the system, and it needs to be clarified it.

Response: Thank you for your comment and suggestion regarding the oxidation half reaction in our study. In the absence of a sacrificial agent, the oxidation reaction can manifest in two scenarios. Firstly, water oxidation can occur where photogenerated holes oxidize water to produce oxygen. Secondly, metal sulfides can undergo photocorrosion, which is currently a significant concern. Specifically, photogenerated holes react with S²⁻ ions present in metal sulfides, leading to the formation of species such as SO₃²⁻ and SO₄^{2-1,2}. However, in our study, we did not detect the generation of oxygen as a product of water oxidation. This suggests that the oxidation reaction observed is more likely attributed to the photocorrosion of metal sulfides. Through a comparison of XPS spectra before and after the catalyst reaction, we observed that the XPS spectrum of the S element after the reaction exhibited a doublet attributed to lattice S²⁻ and a very faint XPS peak corresponding to the SO₃²⁻ species, as shown in **Supplementary Fig. 15d**. For the reacted

catalysts, the binding energies of 161.65 eV and 162.75 eV were assigned to $S2p_{3/2}$ and $S2p_{1/2}$, respectively, while the appearance of new peaks within the range of 168.26–170.26 eV was associated with the XPS peaks of the SO_3^{2-} species^{3,4}. This observation indicates the occurrence of an oxidation reaction within the $CuInSnS_4$ photocatalyst, specifically, the occurrence of photocorrosion. Simultaneously, **Supplementary Fig. 16** shows the O1s spectra of the $CuInSnS_4$ photocatalyst before and after the reaction. For the catalyst before the reaction, a set of peaks with binding energies of 531.77 and 533.18 eV is assigned to surface hydroxyl groups (-OH) and surface adsorbed oxygen (O_2 , CO_3^{2-}), respectively⁵⁻⁷. For the after reaction catalyst, a set of peaks with binding energies of 530.82, 532.09, and 533.20 eV is assigned to lattice oxygen, surface hydroxyl groups, and surface adsorbed oxygen, respectively. It is noted that after the reaction, the binding energy is 530.82 eV similar to lattice oxygen of In_2O_3 (530.76 eV)⁸. Compared to the sample before the reaction, the content of hydroxyl groups on the surface of the catalyst after the reaction remains basically unchanged, but the binding energy increases. Meanwhile, the adsorbed oxygen species on the surface of the catalyst decrease significantly after the reaction. This may be due to the interaction between adsorbed oxygen species and photogenerated carriers that generates active oxygen species to further oxidize S^{2-} to SO_3^{2-} . Therefore, in the photocatalytic oxidation reaction, the photogenerated holes mainly oxidize the adsorbed oxygen and other species adsorbed on the catalyst surface to generate active species and then oxidize the catalyst surface S^{2-} to SO_3^{2-} .

Supplementary Fig. 15 XPS spectra of CuInSnS₄ (180°C) before and after the reaction. (a) In 3d, (b) Cu 2p, (c) Sn 3d, and (d) S 2p.

Supplementary Fig. 16 O 1s spectra of CuInSnS₄ (180°C) photocatalyst before and after reaction.

References

1. Weng, B., Qi, M. Y., Han, C., Tang, Z. R., & Xu, Y. J. Photocorrosion inhibition of semiconductor-based photocatalysts: basic principle, current development, and future perspective. *ACS Catal.* **9**, 4642–4687 (2019). <https://doi.org/10.1021/acscatal.9b00313>
2. Yu, H., Huang, X., Wang, & P., Yu, J. Enhanced photoinduced stability and photocatalytic activity of CdS by dual amorphous cocatalysts: synergistic effect of Ti(IV)-hole cocatalyst and Ni(II)-electron cocatalyst. *J. Phys. Chem. C* **120**, 3722–3730 (2016). <https://doi.org/10.1021/acs.jpcc.6b00126>
3. Weide, P., Schulz, K., Kaluza, S., Rohe, M., Beranek, R., & Muhler, M. Controlling the photocorrosion of zinc sulfide nanoparticles in water by doping with chloride and cobalt ions. *Langmuir*, **32**, 12641–12649 (2016). <https://doi.org/10.1021/acs.langmuir.6b03385>
4. Ding, Y., Chen, Y., Guan, Z., Zhao, Y., Lin, J., Jiao, Y., & Tian, G. Hierarchical CuS@ZnIn₂S₄ hollow double-shelled p–n heterojunction octahedra decorated with fullerene C₆₀ for remarkable selectivity and activity of CO₂ photoreduction into CH₄. *ACS Appl. Mater. Interfaces*, **14**, 7888–7899 (2022). <https://doi.org/10.1021/acscami.1c20980>
5. Jiang, F., Wang, S., Liu, B., Liu, J., Wang, L., Xiao, Y., Xu, Y., & X. Liu. Insights into the Influence of CeO₂ Crystal Facet on CO₂ Hydrogenation to Methanol over Pd/CeO₂ Catalysts. *ACS Catal.* **10**, 11493–11509 (2020). <https://doi.org/10.1021/acscatal.0c03324>
6. Wang, N., Li, S., Zong, Y., & Yao, Q. Sintering inhibition of flame-made Pd/CeO₂ nanocatalyst for low-temperature methane combustion. *J. Aerosol Sci.* **105**, 64–72 (2017). <https://doi.org/10.1016/j.jaerosci.2016.11.017>
7. Wang, B., Chen, B., Sun, Y., Xiao, H., Xu, X., Fu, M., Wu, J., Chen, L., & Ye, D. Effects of dielectric barrier discharge plasma on the catalytic activity of Pt/CeO₂ catalysts. *Appl. Catal., B.* **238**, 328–338 (2018). <https://doi.org/10.1016/j.apcatb.2018.07.044>
8. Sabri, M. M., Jung, J., Yoon, D. H., Yoon, S., Tak, Y. J., & Kim, H. J. Hydroxyl radical-assisted decomposition and oxidation in solution-processed indium oxide thin-film transistors. *J. Mater. Chem. C* **3**, 7499–7505 (2015). <https://doi.org/10.1039/C5TC01457C>

REVIEWER COMMENTS

Reviewer #1 (Remarks to the Author):

Zhang et al. demonstrate the reduction products can be switched from CO to CH₄ by adjusting the metal to the non-metal S atom in metallic sulfides. Indeed, it can be seen that CuInSnS₄ has a multiple increase in selectivity and yield for CH₄ compared to single metal sulfides In₂S₃, Cu₂S, and SnS₂. But the CH₄ production of CuInSnS₄ is very low, only 5.83 $\mu\text{mol h}^{-1} \text{g}^{-1}$. CH₄ production was significantly improved by modifying with cocatalysts such as Pt, CoO_x, NiO_x, and Co (OH)₂, especially Co (OH)₂, which increases its production by nearly 30 folds. The high yield claimed by the authors is largely based on the introduction of cocatalysts. Unfortunately, the synthesis, characterization, and catalytic experiments of composite materials did not contained in the main text or SI, only promoted separation of photo generated electrons and holes is described without any supporting data. We all know that composite strategies can significantly affect photocatalysis, either the type of substance introduced, amount, or synthetic procedure, but these information is missing in the article. The data presented in the article does not provide readers with a clear understanding of this work, Therefore, it is not recommended for acceptance.

Reviewer #2 (Remarks to the Author):

The authors have well addressed our suggestions and the whole manuscript has been improved according to the comments. I am pleased to recommend it for publication.

Reviewer #3 (Remarks to the Author):

It is clear that Zhang's team has worked hard to address all my concerns and I recommend accepting it

Point-by-point response to the reviewers #1

Reviewer #1 (Remarks to the Author):

Zhang et al. demonstrate the reduction products can be switched from CO to CH₄ by adjusting the metal to the non-metal S atom in metallic sulfides. Indeed, it can be seen that CuInSnS₄ has a multiple increase in selectivity and yield for CH₄ compared to single metal sulfides In₂S₃, Cu₂S, and SnS₂. But the CH₄ production of CuInSnS₄ is very low, only 5.83 μmol h⁻¹ g⁻¹. CH₄ production was significantly improved by modifying with cocatalysts such as Pt, CoO_x, NiO_x, and Co(OH)₂, especially Co(OH)₂, which increases its production by nearly 30 folds. The high yield claimed by the authors is largely based on the introduction of cocatalysts. Unfortunately, the synthesis, characterization, and catalytic experiments of composite materials did not contain in the main text or SI, only promoted separation of photo generated electrons and holes is described without any supporting data. We all know that composite strategies can significantly affect photocatalysis, either the type of substance introduced, amount, or synthetic procedure, but these information is missing in the article. The data presented in the article does not provide readers with a clear understanding of this work, Therefore, it is not recommended for acceptance.

Response: Thank you for your helpful comments to improve our manuscript. We are sorry for our omission of the detail information about the synthesis, characterization, and catalytic experiments of the modified CuInSnS₄ in manuscript. We have now supplemented the synthesis processes of the modified CuInSnS₄ in the supporting information. The chemical states of the Pt, CoO, NiO, and Co(OH)₂ co-catalysts were verified through XPS characterization, while the crystal phase and composition of the cocatalyst were evaluated by XRD. We conducted photocurrent and electrochemical impedance tests to investigate the separation efficiency of photogenerated electrons and holes. The results confirmed that the improvement of the separation of photogenerated charge carriers by co-catalysts was consistent with their photocatalytic performance over the modified CuInSnS₄. The above results have been incorporated into the revised manuscript. We sincerely hope that our revised manuscript has adequately addressed your concerns and made it more clearly to

readers.

The specific modification details and results are as follows:

(1) The synthesis processes of the modified CuInSnS₄ photocatalysts

(I) Preparation of 1%Pt/CuInSnS₄ photocatalyst

The Pt modified photocatalyst was prepared by the photodeposition method. In detail, 200 mg of CuInSnS₄ was firstly dispersed in a mixed solution of 150 mL H₂O and 20 mL CH₃OH. And then, 530 μ L H₂PtCl₆·6H₂O solution (10 mg mL⁻¹) was added to the above dispersion liquid. The reaction solution system was evacuated and stirred for 1 hour. Subsequently, visible light was irradiated for 1 hour to reduce H₂PtCl₆·6H₂O to Pt on CuInSnS₄ surface. Finally, the catalyst samples were washed with deionized water by centrifugation and dried under vacuum at 60°C to obtain 1%Pt/CuInSnS₄ photocatalyst.

(II) Preparation of 10%CoO or 10%NiO modified CuInSnS₄ photocatalyst

50 mg CuInSnS₄ was dispersed in 10 mL of H₂O, and then 6.0 mg Co(NO₃)₂·6H₂O or 19.5 mg of Ni(NO₃)₂·6H₂O was added for 3 hour stirring. The H₂O was evaporated at 90°C in an oven. The resulting powder was ground and transferred to a tube furnace for heating treatment at 350 °C for 1 hour with a heating rate of 2 °C min⁻¹ under inert Ar gas atmosphere. After cooling to room temperature, the catalyst was washed several times with deionized water and dried under vacuum at 60°C.

(III) Preparation of Co(OH)₂/CuInSnS₄ photocatalyst

The Co(OH)₂ modified photocatalyst was prepared by the precipitation method. 100 mg of CuInSnS₄ was dispersed in 50 mL of deionized water and stirred evenly. Subsequently, a certain amount of Co(NO₃)₂·6H₂O solution (1 mg·mL⁻¹) was added, and stirring was continued for 12 hours. Finally, 10 mL of NaOH solution (2 mg mL⁻¹) was added, and stirring was continued for an additional 12 hours. After the reaction, the samples were washed several times with deionized water and dried under vacuum. The amounts of Co(NO₃)₂·6H₂O solution addition were 9.39, 15.66, 25.06 and 31.32 mL, corresponding to 3%, 5%, 8% and 10% Co(OH)₂ loading to CuInSnS₄, respectively.

(IV) Preparation of 5%Co(OH)₂/CuInSnS₄/1%Pt photocatalyst

The Co(OH)₂/CuInSnS₄/Pt photocatalyst was prepared in two steps. Firstly, 1%Pt was deposited on the surface of the CuInSnS₄ sample through photodeposition. Subsequently, 5%Co(OH)₂ was loaded on the surface of Pt/CuInSnS₄ by the precipitation method.

(2) Characterization of CuInSnS₄ photocatalysts modified with a series of co-catalysts

(I) Crystal phase structure analysis

Supplementary Fig. 18 XRD patterns of (a) 5%Co(OH)₂/CuInSnS₄, (b) 5%Co(OH)₂/CuInSnS₄/1%Pt, (c) 10%CoO/CuInSnS₄, and (d) 10%NiO/CuInSnS₄.

The XRD patterns determined the crystal phase of CuInSnS₄ samples modified with a series of co-catalysts, as presented in **Supplementary Fig. 18**. In the case of 5%Co(OH)₂-modified CuInSnS₄ photocatalysts, as well as 5%Co(OH)₂ and 1%Pt co-modified CuInSnS₄ sample, the XRD patterns exhibited only the diffraction peaks corresponding to cubic CuInSnS₄. No distinct diffraction peaks for Co(OH)₂ or Pt species were observed. The absence of the diffraction peaks for Pt species could possibly be attributed to the low concentration. Additionally, the lack of clear diffraction peaks for

Co(OH)₂ might arise from its relatively low crystallinity. In contrast, for NiO and CoO cocatalyst-modified CuInSnS₄ photocatalysts, the XRD analysis successfully detected diffraction peaks attributed to CoO and NiO. Both CoO and NiO co-catalysts exhibited patterns consistent with cubic CoO (PDF#48-1719) and NiO (PDF#02-1216).

(II) XPS characterization of cocatalysts

(a) 5%Co(OH)₂/CuInSnS₄ sample

Supplementary Fig. 19 (a) Co 2p and (b) O 1s XPS spectra of 5%Co(OH)₂/CuInSnS₄ sample.

XPS was employed to confirm the chemical states of Co(OH)₂ cocatalyst. **Supplementary Figure 19a** illustrated the Co 2p XPS spectra in 5%Co(OH)₂/CuInSnS₄ sample. A set of peaks with binding energies of 781.27 and 797.42 eV corresponded to Co2p_{3/2} and Co2p_{1/2} of Co(OH)₂, respectively. It was noteworthy that the binding energy difference ($\Delta = 16.1$ eV) between these two peaks proved the presence of Co in the form of Co(OH)₂, in accordance with findings from literature reports ^{1,2}. Furthermore, an additional set of peaks with binding energies of 785.72 and 803.29 eV represented the satellite peaks of Co²⁺ of Co(OH)₂ cocatalyst ³. Additionally, O 1s XPS spectra also revealed three peaks with binding energies of 531.25, 532.29, and 533.39 eV, which were ascribed to lattice oxygen of Co(OH)₂, absorbed hydroxyl, and surface absorbed H₂O or O₂, respectively ^{2,4,5} (**Supplementary Fig. 19b**). Therefore, Co(OH)₂ co-catalyst was successfully modified on the surface of CuInSnS₄ photocatalyst.

(b) 10%NiO/CuInSnS₄ sample

The chemical states of NiO cocatalyst was analyzed and confirmed by XPS. **Supplementary Fig. 20a** showed XPS spectra of the Ni 2p in NiO/CuInSnS₄ sample. A group of peaks with binding energies of 855.50 and 873.68 eV corresponded to Ni2p_{3/2} and Ni2p_{5/2} of Ni²⁺, respectively ^{6,7}. Another group of peaks with binding energies of 861.81 and 880.35 eV represented the satellite peaks of Ni²⁺ ⁸. An obvious multiple splitting phenomenon in the main XPS peak of Ni2p_{3/2} was consistent with NiO. Hence, it can be inferred that NiO was successfully supported on the surface of CuInSnS₄ sample. **Supplementary Fig. 20b** of O 1s XPS spectra showed that the three peaks with binding energies of 530.15, 531.83, and 533.01 eV were assigned to lattice oxygen of NiO, surface hydroxyl, and surface adsorbed oxygen, respectively ^{4,5}. Notably, for pure CuInSnS₄, only the XPS peaks of surface hydroxyl groups and surface adsorbed oxygen at binding energies of 531.77 and 533.15 eV were detected. Hence, the XPS peak at 530.15 eV was assigned to the lattice oxygen of NiO ⁶.

Supplementary Fig. 20 (a) Ni 2p and (b) O1s XPS spectra of 10%NiO/CuInSnS₄ sample.

(c) 10%CoO/CuInSnS₄ sample

The chemical states of the CoO cocatalyst were analyzed and verified by XPS. **Supplementary Fig. 21a** presented XPS spectra of Co 2p in CoO/CuInSnS₄ sample. The Co 2p XPS peaks at 780.76 eV (Co 2p_{3/2}) and 796.80 eV (Co 2p_{1/2}) revealed distinct characteristic peaks of the Co²⁺ oxidation state ⁹. Moreover, their satellite peaks appeared at binding energies of 785.25 and 802.89 eV further confirming the presence of CoO. **Supplementary Fig. 21b** of O 1s XPS spectra showed that the three

peaks with binding energies of 530.53, 531.76, and 532.91 eV were assigned to lattice oxygen of CoO, surface hydroxyl, and surface adsorbed oxygen, respectively ^{4,5}.

Supplementary Fig. 21 (a) Co 2p and (b) O 1s XPS spectra of 10%CoO/CuInSnS₄ sample.

(d) 1%Pt/CuInSnS₄ sample

Supplementary Fig. 22 Pt 4f XPS spectra of 1%Pt/CuInSnS₄ sample.

Supplementary Fig. 22 showed Pt 4f XPS spectra of 1%Pt/CuInSnS₄ sample. A set of peaks with binding energies of 70.59 and 73.92 eV was assigned to Pt4f_{7/2} and Pt4f_{5/2} of Pt⁰, respectively, while

another set of peaks with binding energies of 72.72 and 75.50 eV was attributed to Pt4f_{7/2} and Pt4f_{5/2} of Pt²⁺, respectively ¹⁰. Evidently, Pt primarily existed on the surface of CuInSnS₄ sample in the form of metal Pt⁰, accompanied by a part of PtO.

(e) 5%Co(OH)₂/CuInSnS₄/1%Pt sample

XPS was employed for the chemical state of Co(OH)₂ and Pt cocatalysts in 5%Co(OH)₂/CuInSnS₄/1%Pt sample, as shown in the **Supplementary Fig. 23**. The two characteristic peaks with binding energies of 70.77 and 74.19 eV were assigned to Pt 4f_{7/2} and Pt 4f_{5/2} of Pt⁰, respectively, while the two characteristic peaks with binding energies of 72.98 and 75.80 eV were assigned to Pt 4f_{7/2} and Pt 4f_{5/2} of Pt²⁺, respectively ¹⁰ (**Supplementary Fig. 23a**). As depicted in **Supplementary Fig. 23b**, the Co 2p spectrum exhibited two characteristic peaks with binding energies of 781.13 and 797.15 eV, corresponding to Co 2p_{3/2} and Co 2p_{1/2} ^{1,2}. The position and distance ($\Delta=16.02$ eV) between these two peaks confirmed the existence of Co in the form of Co(OH)₂. The two characteristic peaks with binding energies of 786.49 and 803.56 eV represented the characteristic satellite peaks of Co²⁺ ³. Therefore, it can be determined that the Co(OH)₂ and Pt cocatalysts were successfully deposited on the surface of CuInSnS₄ sample.

Supplementary Fig. 23 (a) Pt 4f and (b) Co 2p XPS spectra of 5%Co(OH)₂/CuInSnS₄/1%Pt.

(3) Separation efficiency of photogenerated carriers

Supplementary Fig. 24 Photocurrent response and electrochemical impedance spectroscopy of the as-prepared samples.

The modification of cocatalysts significantly enhanced the photocatalytic activity of polymetallic sulfide CuInSnS_4 nanosingle crystals. Photoelectrochemical characterization was employed to investigate the separation efficiency of photogenerated carriers (**Supplementary Fig. 24**). All of cocatalysts modified CuInSnS_4 photocatalysts displayed higher photocurrent signals and smaller electrochemical impedance radii compared to pure CuInSnS_4 samples. Therefore, the modification of cocatalysts can enhance the separation and migration of photogenerated charges of CuInSnS_4 . The photocurrent increased sequentially in the order of $\text{CuInSnS}_4 < 5\% \text{Co(OH)}_2/\text{CuInSnS}_4 \approx 1\% \text{Pt}/\text{CuInSnS}_4 < 5\% \text{Co(OH)}_2/\text{CuInSnS}_4/1\% \text{Pt}$, indicating that the modification of the dual co-catalyst further improved photoelectric carrier separation and migration compared to the single co-catalyst. Furthermore, the modified NiO and CoO cocatalysts also correspondingly improved the separation efficiency and migration rate of the photogenerated carriers in the CuInSnS_4 photocatalyst. The decreasing order of electrochemical resistance radius was $\text{CuInSnS}_4 > 1\% \text{Pt}/\text{CuInSnS}_4 \approx 5\% \text{Co(OH)}_2/\text{CuInSnS}_4 > 5\% \text{Co(OH)}_2/\text{CuInSnS}_4/1\% \text{Pt}$, consistent with the activity trend. Additionally, both $10\% \text{NiO}/\text{CuInSnS}_4$ and $10\% \text{CoO}/\text{CuInSnS}_4$ exhibited smaller electrochemical resistance radii than pure CuInSnS_4 , further confirming that the cocatalyst promoted the photogenerated charge

separation and migration. This enhancement significantly contributed to the improved photocatalytic CO₂ reduction activity.

(4) Photocatalytic CO₂ reduction performance

Although the pristine CuInSnS₄ only exhibited a yield of CH₄ evolution of 6.53 μL h⁻¹ (corresponding to 5.83 μmol h⁻¹ g⁻¹) with a selectivity of 67.3%, the activity and selectivity of CH₄ evolution on CuInSnS₄ can be improved by coupling with other semiconductor-based photocatalysts or noble metals as cocatalysts. We evaluated the photocatalytic performance of CuInSnS₄ modified with Pt, CoO, NiO, Co(OH)₂ and dual co-catalysts Pt and Co(OH)₂, for the CO₂ reduction reaction. Table S2 listed a comparison of the photoreduction activity of CuInSnS₄ and co-catalyst modified CuInSnS₄ photocatalysts, along with the common metal sulfide systems currently used for CO₂ photoreduction. Clearly, both the yield and selectivity of CH₄ evolution on CuInSnS₄ can be significantly improved by modifying with cocatalysts such as Pt, CoO, NiO, and Co(OH)₂. The activity of CH₄ evolution on the modified CuInSnS₄ photocatalysts surpassed the majority of the reported photocatalysts for CO₂ reduction up to now. Particularly, the incorporation of Co(OH)₂ as a co-catalyst significantly enhanced the CO₂ photoreduction activity of the CuInSnS₄ photocatalyst. As the Co(OH)₂ loading increases, the photoreduction activity of CO₂ exhibited a characteristic volcanic pattern. With 5%Co(OH)₂ loading onto CuInSnS₄, the production rates for CH₄ and CO respectively reached 145.45 and 32.32 μmol h⁻¹ g⁻¹, corresponding to a CH₄ selectivity of 81.8%. The generation rates of CH₄ and CO were 25 times and 13 times that of pure CuInSnS₄, respectively. Furthermore, when CuInSnS₄ was modified with a dual co-catalyst of 5%Co(OH)₂ as an oxidation co-catalyst and 1%Pt as a reduction co-catalyst, CH₄ production reached 195.60 μmol h⁻¹ g⁻¹, along with 22.00 μmol h⁻¹ g⁻¹ of CO, and a CH₄ selectivity of 89.9%. Therefore, the modification of CuInSnS₄ with various cocatalysts to enhance the separation of photogenerated carriers can effectively improve the activity and selectivity of CH₄ products.

Table S2 The photocatalytic CO₂ reduction performance of CuInSnS₄, and modified CuInSnS₄ photocatalysts.

Catalyst	Generation rate of CH ₄ (μmol h ⁻¹ g ⁻¹)	Generation rate of CO (μmol h ⁻¹ g ⁻¹)	Selectivity of CH ₄	Ref.
CuInSnS ₄	5.83	2.40	67.3%	This work
1%Pt/CuInSnS ₄	43.25	7.85	84.6%	This work
10%CoO/CuInSnS ₄	33.30	9.36	78.1%	This work
10%NiO/CuInSnS ₄	11.80	6.85	63.3%	This work
3%Co(OH) ₂ /CuInSnS ₄	18.50	3.02	86.0%	This work
5%Co(OH) ₂ /CuInSnS ₄	145.45	32.32	81.8%	This work
8%Co(OH) ₂ /CuInSnS ₄	46.80	13.22	78.0%	This work
10%Co(OH) ₂ /CuInSnS ₄	18.30	6.20	74.7%	This work
5%Co(OH) ₂ /CuInSnS ₄ /1%Pt	195.60	22.00	89.9%	This work
Mn(II)-metalated porphyrin	~53	~21	71.2%	17
CuS@ZnIn ₂ S ₄ /C ₆₀	43.6	6.4	87.2%	18
Pt@h-BN	9.24	0	100%	19
SnS ₂ /TiO ₂	23	2.5	90.2%	20
SiC@MoS ₂	14.42	0	100%	21

V _o -Zn-CoO	17.10	9.70	63.8%	22
In ₄ SnS ₈	23.88	20.96	57.0%	23
V _S -CuIn ₅ S ₈ single-unit-cell layers	8.70	0	100%	24
Cu ₂ O-111-Cu ⁰	78.40	0	97%	25
H-TiO _{2-x} (200)	16.20	4.2	79%	26
Ag ₄ /Cu ₂ O@rGO	82.6	4.0	95.4%	27
Cs ₂ SnI ₆ /SnS ₂	~6.09	0	~100%	28
Pt-Cu ₂ O/TiO ₂	1.42	0.05	96.6%	29
N-RGO/CdS	0.33	2.59	11.3%	30
Au _{SA} /Cd _{1-x} S	11.3	32.2	22.0%	31
Au _{SA} /CdS _{1-x}	0.40	3.70	9.3%	31
Ni-doped CoS ₂ nanosheets	101.8	37.5	~73.10%	32
Pd ₇ Cu ₁ -TiO ₂	19.60	1.90	95.9%	33
Ag ₂ Pd ₁ /TiO ₂	79.0	0	100%	34

References

1. Yang, J., Liu, H., Martens, W. N., & Frost, R. L. Synthesis and characterization of cobalt hydroxide, cobalt oxyhydroxide, and cobalt oxide nanodiscs. *J. Phys. Chem. C* **114**, 111–119 (2010).
2. Luo, Y., Li, X., Cai, X., Zou, X., Kang, F., Cheng, H.-M., & Liu, B. Two-dimensional MoS₂ confined Co(OH)₂ electrocatalysts for hydrogen evolution in alkaline electrolytes. *ACS Nano* **12**, 4565–4573 (2018).
3. Pal, D., Sarkar, A., Ghosh, N. G., Sanke, D. M., Maity, D., Karmakar, K., Sarkar, D., Zade, S. S., & Khan, G. G. Integration of LaCo(OH)_x photo-Electrocatalyst and plasmonic gold nanoparticles with Sb-doped TiO₂ nanorods for photoelectrochemical water oxidation. *ACS Appl. Nano Mater.* **4**, 6111–6123 (2021).
4. Jiang, F., Wang, S., Liu, B., Liu, J., Wang, L., Xiao, Y., Xu, Y., & X. Liu. Insights into the Influence of CeO₂ Crystal Facet on CO₂ Hydrogenation to Methanol over Pd/CeO₂ Catalysts. *ACS Catal.* **10**, 11493–11509 (2020).
5. Wang, B., Chen, B., Sun, Y., Xiao, H., Xu, X., Fu, M., Wu, J., Chen, L., & Ye, D. Effects of dielectric barrier discharge plasma on the catalytic activity of Pt/CeO₂ catalysts. *Appl. Catal., B.* **238**, 328–338 (2018).
6. Zhang, P.-F., Zhang, J.-Y., Sheng, T., Lu, Y.-Q., Yin, Z.-W., Li, Y.-Y., Peng, X.-X., Zhou, Y., Li, J.-T., Wu, Y.-J., Lin, J.-X., Xu, B.-B., Qu, X.-M., Huang, L., & Sun, S.-G. Synergetic effect of Ru and NiO in the electrocatalytic decomposition of Li₂CO₃ to enhance the performance of a Li-CO₂/O₂ battery. *ACS Catal.* **10**, 1640–1651 (2020).
7. Merum, D., Nallapureddy, R., R., Pallavolu, M. R., Mandal, T. K., Gutturu, R. R., Parvin, N., Banerjee, A. N., & Joo, S. W. Pseudocapacitive performance of freestanding Ni₃V₂O₈ nanosheets for high energy and power density asymmetric supercapacitors. *ACS Appl. Energy Mater.* **5**, 5561–5578 (2022).
8. Chen, G., Chen, D., Huang, J., Zhang, C., Chen, W., Li, T., Huang, B., Shao, T., Li, J., & Ostrikov, K. K. Focused plasma-and pure water-enabled, electrode-emerged nanointerfaced NiCo hydroxide–oxide for robust overall water splitting. *ACS Appl. Mater. Interfaces* **13**, 45566–45577 (2021).
9. Li, R., Rao, D., Zhou, J., Wu, G., Wang, G., Zhu, Z., Han, X., Sun, R., Li, H., Wang, C., Yan, W.,

Zheng, X., Cui, P., Wu, Y., Wang, G., & Hong, X. Amorphization-induced surface electronic states modulation of cobaltous oxide nanosheets for lithium-sulfur batteries. *Nat Commun* **12**, 3102 (2021).

10. Hatanaka, M. et al. Reversible changes in the Pt oxidation state and nanostructure on a ceria-based supported Pt. *J. Catal.* **266**, 182–190 (2009).